# GRADE: QUANTIFYING SAMPLE DIVERSITY IN TEXT-TO-IMAGE MODELS

## ABSTRACT

Text-to-image (T2I) models are remarkable at generating realistic images based on textual descriptions. However, textual prompts are inherently *underspecified*: they do not specify all possible attributes of the required image. This raises two key questions: Do T2I models generate diverse outputs from underspecified prompts? How can we automatically measure diversity? We propose GRADE: **Gr**anular **A**ttribute **D**iversity **E**valuation, an automatic method for quantifying sample diversity. GRADE leverages the world knowledge embedded in large language models and visual question-answering systems to identify relevant concept-specific axes of diversity (e.g., "shape" and "color" for the concept "cookie"). It then estimates frequency distributions of concepts and their attributes and quantifies diversity using (normalized) entropy. GRADE achieves over 90% human agreement while exhibiting weak correlation to commonly used diversity metrics. We use GRADE to measure the overall diversity of 12 T2I models using 405 concept-attribute pairs, revealing that all models display limited variation. Further, we find that these models often exhibit *default behaviors*, a phenomenon where the model consistently generates concepts with the same attributes (e.g., 98% of the cookies are round). Finally, we demonstrate that a key reason for low diversity is due to underspecified captions in training data. Our work proposes a modern, semantically-driven approach to measure sample diversity and highlights the stunning homogeneity in outputs by T2I models.

## 1 INTRODUCTION

Text-to-image (T2I) models have the remarkable ability to generate realistic images based on textual descriptions. However, prompts are inherently *underspecified* (Hutchinson et al., 2022; Rassin et al., 2022), meaning they do not fully describe all attributes that appear in the resulting image. Often, we expect T2I models to *produce diverse outputs* that represent the full spectrum of possible attributes. For example, when generating images of "a cookie in a bakery", we expect to see cookies with different shapes, colors, and textures, among other variations. But are current T2I models capable of generating diverse outputs? Evaluating diversity is inherently challenging because the set of possible attributes is virtually infinite. Existing metrics, such as Fréchet Inception Distance (FID) (Heusel et al., 2017) and Precision-and-Recall (Sajjadi et al., 2018; Kynkäänniemi et al., 2019) are supposed to measure diversity, but they are limited in their ability to capture granular forms of diversity, instead, they capture feature-level similarities. These metrics also rely on a set of reference images that typically reflects the training data distribution, which might not be diverse. Furthermore, such set is often hard to obtain, and does not specify attributes of interest. Our desiderata from a diversity metric is to be reference-free, independent of the training data distribution, and human-interpretable.

We propose **Gr**anular **A**ttribute **D**iversity **E**valuation (GRADE), a method for measuring sample diversity in T2I models at a granular, **concept-dependent** manner, focusing on attributes of concepts, such as the *shape* of a *cookie* or the *state* of an *umbrella*. Our approach (illustrated in Fig. 2) involves using a large language model (LLM) to generate prompts that elicit diverse outputs from T2I models. These prompts are accompanied by questions that tailor common, specific *attributes*–relevant axes of diversity–for each concept (e.g., "What is the shape of the cookie?" and "Is the umbrella open or close?"). We use a visual question-answering (VQA) model to extract attribute values from images using the questions. We then use an LLM to approximate the support of the concept and

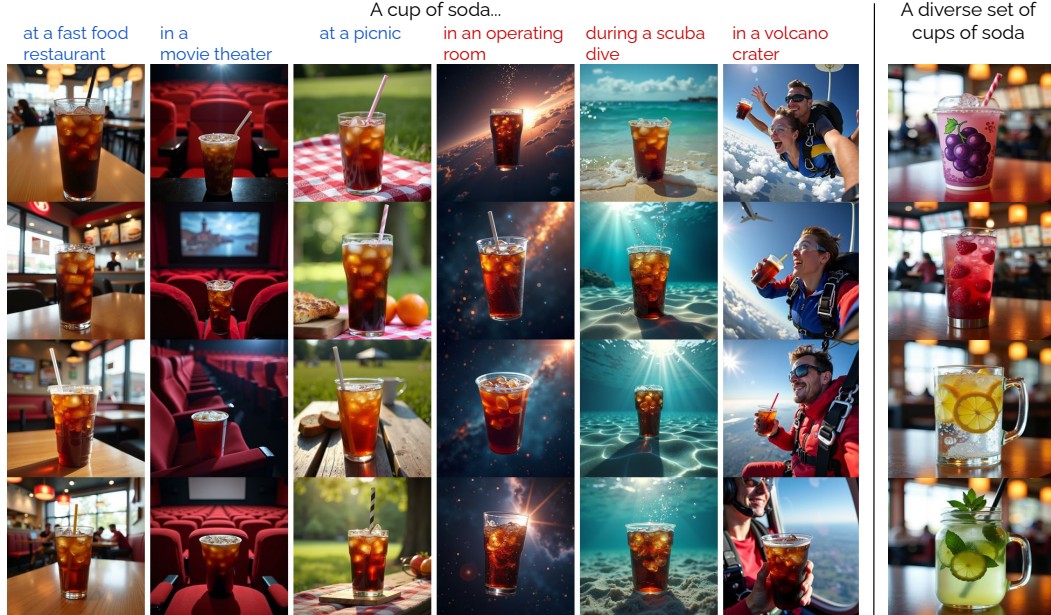

A cup of soda...

at a fast food restaurant | in a movie theater | at a picnic | in an operating room | during a scuba dive | in a volcano crater

A diverse set of cups of soda

Figure 1: A sample of four images per prompt. The model is not diverse: all soda cups contain cola with ice, though the input prompt for "cup of soda" is underspecified. The rightmost column demonstrates a set of diverse cups, not achievable without directly specifying in the prompt. All images are outputs of FLUX.1-dev.

attribute, and map the VQA outputs to attribute values in the support. The result is a distribution over a concept and an attribute. We compute its normalized entropy and use it as our diversity score.

Using GRADE, we determine that no model we test is particularly diverse, with the highest diversity score being 0.64 on a scale from zero to one. For instance, the generated images of "cup of soda" by FLUX.1-dev (shown in Fig. 1), the state-of-the-art T2I model, exhibit extremely low entropy (0.1 bits), and are round 96% of the time, a phenomenon we name *default behavior*. We explain such low scores from non-diverse images in the training data, that often appear with underspecified captions, which was previously explored in societal biases associations Seshadri et al. (2023).

Our contributions are threefold:

- **A novel diversity evaluation method:** We introduce GRADE, a fine-grained and interpretable method for evaluating diversity in T2I models that does not rely on reference images. We show GRADE captures forms of diversity FID and Recall do not, even with the presence of reference images.

- **Comparative diversity analysis:** Using GRADE, we conduct an extensive study comparing the diversity of 12 T2I models, revealing that even the most diverse ones achieve low diversity and frequently exhibit *default behaviors*. Our analysis uncovers negative correlation between model size and diversity.

- **Insights into influence of training data:** We demonstrate that underspecified captions in the training data contribute to low diversity of underspecified prompts.

## 2 RELATED WORK

Most diversity measurements are *distribution-based*: a set of images generated by the evaluated model is compared to a reference set that captures the desired diversity, typically in feature-space, using a feature extractor such as Inception v3 (Szegedy et al., 2014; Salimans et al., 2016) or CLIP (Radford et al., 2021).

Perhaps most popular, Fréchet Inception Distance (FID) outputs a score representing both fidelity and diversity and is the standard for evaluating image generating models. However, it has multiple documented issues, like numerical sensitivity, data contamination, and biases (Parmar et al., 2022; Bińkowski et al., 2018; Chong & Forsyth, 2020; Kynkäänniemi et al., 2022; Jayasumana et al., 2024). Precision-and-Recall (Sajjadi et al., 2018) separated fidelity and diversity to two metrics. Additional metrics were proposed Kynkäänniemi et al. (2019); Naeem et al. (2020); Kim et al. (2023); Alaa et al. (2022), which decouple between different properties and offer more interpretable methods. Crucially, all these methods rely on a set of **diverse** reference images, by comparing the distribution of generated images to the reference set with the desired level of diversity. This can be the model's training data, or an established dataset, like ImageNet (Deng et al., 2009). However, acquiring reference images that faithfully reflect diversity is not straightforward and often requires using a feature extractor that was trained on similar data, to capture the similarities between the distributions. These requirements make it difficult to reproduce the results of previous work and maintain the integrity of the metrics as they are sensitive to data contamination, which could make them favor models that produce patterns similar to those seen in their training set, regardless of diversity (Kynkäänniemi et al., 2022).

In addition to the significant requirement of obtaining a feature extractor and training data that match the target domain, previous metrics do not use fine-grained feature extractors, which can evaluate diversity over the semantics of images. Instead, they use ones that are trained over well-established datasets. As a result, they lack the ability to distinguish between two similar concepts that are different on a specific axis, like a color. For example, if we compare two nearly-identical images of a bottle, with only the color of the bottle as the difference, they would consider them very similar. However, our metric would capture such difference, as we show in Appendix C.

Similar to GRADE, Vendi Score (VS) (Friedman & Dieng, 2022; Pasarkar & Dieng, 2023) is a reference-free metric, defined as the entropy of the eigenvalues of a user-provided similarity metric. However, VS is sensitive to the choice of similarity function and like previous approaches, it is not fine-grained and interpretable in natural text.

## 3 GRADE: MEASURING DIVERSITY IN TEXT-TO-IMAGE MODELS

### 3.1 APPROACH

Our goal is to quantify the diversity of generated images over attributes of some concept while its attributes are underspecified in the prompt.

Let $C$ be a random variable representing concepts, taking values $c$ (e.g., "cookie"). Let $A$ be a random variable representing attributes, taking values $a$ (e.g., "shape"). The set of possible attribute values that attribute $a$ can take for a concept $c$ is denoted by $\mathcal{V}_c^a$ (e.g., $\mathcal{V}_{\text{cookie}}^{\text{shape}}$ might include "round", "square", "rectangular", etc.). We define $V$ as a random variable representing the attribute values for attribute $A = a$ and concept $C = c$, taking values $v$ from the set $\mathcal{V}_c^a$.

We define the *concept distribution* $P_{V|a,c}(v)$ as the probability that a generated image of concept $c$ exhibits the attribute value $v$ for attribute $a$:

$$P_{V|a,c}(v) = P(V = v \mid A = a, C = c) \tag{1}$$

Ideally, to obtain $P_{V|a,c}(v)$, one would consider all generatable images of $c$ and count the occurrences of each attribute value $v$. However, this is infeasible due to the vast nature of attribute spaces and the variety of images with concept $c$. Furthermore, the relevant attributes are concept-dependent and rely on world knowledge (e.g., "open or closed" is a relevant distinction for a parachute, but not for a cookie). Thus, we instantiate the attribute values set $\mathcal{V}_c^a$ with an approximation $\tilde{\mathcal{V}}_c^a$, which serves as the support of our approximated distribution and relies on the world knowledge of LLMs. We also define a set of prompts $\mathcal{P} = \{p_1, p_2, \ldots, p_n\}$ that mention the concept $c$ but not $\mathcal{V}_c^a$.

We then estimate the *multi-prompt distribution*, as an approximation of the concept distribution $\tilde{P}_{V|a,c}(v)$ by generating images using the prompts in $\mathcal{P}$ and counting the occurrences of each attribute value $v \in \tilde{\mathcal{V}}_c^a$:

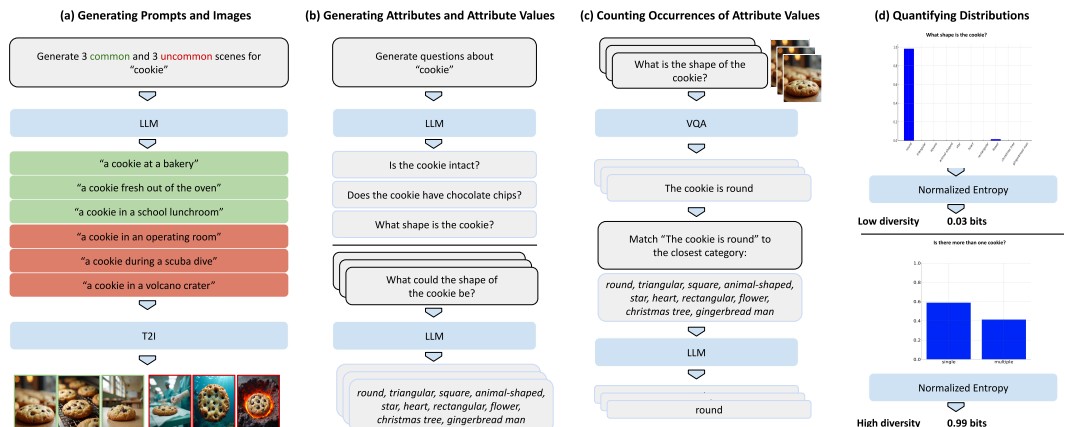

Figure 2: Workflow of GRADE with "cookie" as input concept $c$ and distributions $\tilde{P}_{V|a,c}$ as output: (a) generating prompts that mention $c$, but are underspecified with respect to attributes, and using them to generate images; (b) generating questions pertaining to attributes $A$ and extracting answers from images using a VQA; (c) generating attribute values $\mathcal{V}_c^a$ and mapping answers to them; (d) measuring diversity using entropy.

$$\tilde{P}_{V|a,c}(v) = \frac{1}{n} \sum_{i=1}^{n} P(V = v \mid A = a, C = c, p_i) \tag{2}$$

We compute the normalized entropy $\hat{H}(\tilde{P}_{V|a,c})$ of the multi-prompt distribution:

$$\hat{H}(\tilde{P}_{V|a,c}) = \frac{H(\tilde{P}_{V|a,c})}{\log_2 |\tilde{\mathcal{V}}_c^a|} \tag{3}$$

where $H(\tilde{P}_{V|a,c})$ is the entropy of the estimated distribution $\tilde{P}_{V|a,c}$, and $|\tilde{\mathcal{V}}_c^a|$ is the size of the support (the number of attribute values in $\tilde{\mathcal{V}}_c^a$). The normalized entropy ranges from 0 to 1 where high values indicate high diversity (a uniform distribution), and low values indicate low diversity (the model consistently generates the same attribute value).

While we primarily focus on multi-prompt distributions, we also report results on *single prompt distributions*, which models the relation between $c$ and $v$ for a specific prompt $p \in \mathcal{P}$.

By measuring the normalized entropy (henceforth referred to as entropy) across various concepts and attributes, we obtain a representative measure of the overall diversity of the T2I model. We compute the mean normalized entropy over all evaluated distributions to summarize the model's diversity performance.

## 3.2 METHOD

GRADE measures model diversity in four steps, as shown in Fig. 2. First, an LLM generates prompts, attributes (as questions), and attribute values. Then, it estimates the distribution over concepts and attributes by generating images with the evaluated T2I model and answering questions via a VQA model. Finally, we quantify the diversity of the distribution using entropy. We now describe each of these steps, and then evaluate each one individually (Section 4).

**(a) Generating images of the concept** $c$. To thoroughly assess the diversity of attribute values that T2I models associate with a concept $c$, GRADE generates two types of prompts. *Common prompts* depict the concept in familiar and frequently encountered settings (e.g., "a cookie during Christmas festivities"). They are likely to produce images reflecting common attribute associations learned from the training data (e.g., a tree-shaped cookie). *Uncommon prompts* place the concept in rare contexts (e.g., "a cookie in a volcano crater"). Such prompts are designed to push

the model beyond typical scenarios, potentially revealing attribute associations that are not solely driven by common context cues. By analyzing the attribute values generated from both common and uncommon prompts, we can identify which attributes are consistently associated with the concept across different contexts. This consistency indicates that certain attributes are inherently linked to the concept in the model's internal representation, rather than being influenced mainly by the content of the prompts. For instance, if the model frequently generates round cookies in both common and uncommon contexts, it suggests that "roundness" is a fundamental attribute of "cookie" within the model's learned knowledge.

**(b) Generating attributes and their values.** After generating prompts and images, we look for attributes relevant for a concept $c$. GRADE uses an LLM to first outline the concept's attributes and then generate questions that probe for their values. For example, upon receiving the input "cookie", the LLM noted that cookies can be made in different shapes, and proposed the question "what is the shape of the cookie?". Next, we generate attribute values (a support), which we use in the next step to map VQA outputs to attribute values. We pair prompts with questions, and for each pair, we instruct an LLM to propose attribute values based on the prompt (e.g., what could be the answer to "what is the shape of the cookie?" for an image generated by the prompt "a cookie in a bakery"?). We then use an LLM to unify the attribute values to a set without semantic repetitions (e.g., "round" and "circle" will be unified to a single value in the set $\tilde{\mathcal{V}}_{cookie}^{shape}$. Table 1 shows a sample of concepts, questions, and attribute values.

**(c) Counting occurrences of attribute values in images.** We pair all images and questions associated with the same concept and feed it to a VQA model, which outputs answers in natural language. Then, we provide an LLM the answers and the attribute values $\tilde{\mathcal{V}}_c^a$, and instruct it to match the answer to the closest matching attribute value (e.g., "The cake is round" will be mapped to the "round" value). To address edge cases, where the T2I model fails to depict the concept in the image or that the answer is not covered by the support, we add "none of the above" as an additional attribute value and discard all answers that are matched to it. This step results with frequency distributions which we normalize to cumulatively sum to 1.

**(d) Quantifying distributions.** GRADE outputs probability distributions. Every concept and attribute pair results in a multi-prompt distribution, and every prompt and attribute pair results in a single prompt distribution. We quantify the diversity of these distributions using entropy as our score. While not a direct focus in this work, a diversity score spanning multiple attributes of a concept can be achieved simply by averaging the entropy values from the multi-prompt (or single prompt) distributions of a concept.

**Implementation details.** In step (a) we generate three prompts from each type (six in total) and 100 images per prompt. In step (b) we generate an average of four questions (which represent attributes) per concept. We use GPT-4o (OpenAI et al., 2024) (`gpt-4o-2024-08-06`) in all steps, with max tokens set to $1,000$, and temperature set to $0$. While we present the question answering and attribute value mapping as separate steps, we perform them in one step, using the Structured Outputs feature (OpenAI, 2024). All prompts are detailed in Appendix H.

The cost of estimating a multi-prompt distribution is approximately \$0.75, and a single prompt distribution is \$0.12, achieved through batch inference. In our experience, wait time is several minutes. The Images were generated using a single A100-80GB.

## 4 VALIDATING GRADE

We evaluate each step in GRADE–except step(d), which only involves applying the normalized entropy formula–and find that all steps are highly accurate. We manually validate the quality of all generated prompts, questions, and attribute values and use human annotators to evaluate 2,800 images sampled from the generations of the 12 models. This validation is performed to establish the reliability of GRADE and is not required every time the method is used.

**(a) Prompt validity.** We manually review all 600 generated prompts and verify that they mention their underspecified concept i.e., the prompt mentions the concept but no attributes. To determine their commonness, we extract the nouns from each prompt and assess their co-occurrence in LAION-5B (Schuhmann et al., 2022) using WIMBD (Elazar et al., 2024), a tool designed for efficiently counting and searching large datasets. We find that all prompts mention their respective underspec-

Table 1: Sample of concepts, attributes, and attribute values. Each concept-attribute pair is a multi-prompt distribution. For brevity, we only show one attribute per concept with more examples in Appendix B.

| Concept | Question (Attribute) | Attribute Values |
|---|---|---|
| Bin | What material is the bin made from? | mesh, cardboard, carbon fiber, rubber, wood, bamboo, wicker, plastic, ceramic, stainless steel, fiberglass, metal, aluminum, steel, fabric, glass |
| Person | Does the person appear to be alone or with others? | alone, with others |
| Suitcase | Is this a vintage suitcase? | yes, no |
| Cake | What flavor is the cake? | tiramisu, cheesecake, carrot, chocolate, strawberry, vanilla |
| Pool | Is the pool indoor or outdoor? | indoor pool, outdoor pool |

ified concept and that the average co-occurrence for common prompts is 30,655, compared to 956 for uncommon prompts, which confirms they match their categories. We do not evaluate the quality of the generated images, as they depend on the T2I model, not GRADE.

**(b) Attribute and attribute values validity.** We manually validate that each of the 405 questions are about an attribute that can be measured by viewing an image that depicts the concept. Next, we manually validate that the support does not have duplicate attribute values (e.g., "round" and "circle" in the same support, if they both pertain to cookie shapes). After that, we verify that the support aptly covers the attribute values extracted from the images: we examine all "none of the above" selections from the crowdsourcing evaluation in the next step. Out of the 1,000 examples used for the first evaluation, only 115 were mapped to it. Out of these, only three (2.6%) are because the answer is not reflected in one of the attribute values. 92 times (9.2%), the T2I model did not include the concept mentioned in the prompt–the model did not adhere to the prompt, and in the other 20 (0.2%), the VQA or workers did not answer the question correctly.

**(c) Answerability of the questions.** We validate the ability of GPT-4o to answer the questions generated in step (b) using Amazon Mechanical Turk (AMT) crowdsourcing platform–once to gauge the agreement over all images, and a second time to gauge the robustness of the agreement on a specific concept. Each example includes a question, an image, and the generated support (including the "none of the above" option). The workers are requested to answer the question by selecting the attribute value in the support that best matches the question and image. Each example is provided to three workers, we take the majority decision. First, we run it on a sample of 1,000 examples from 12 T2I models and find that the answers by GPT-4o match the answers by the majority decision for **90.2%** of the cases. Then, we run this step a second time with all 600 images of the "what is the shape of the cake?" multi-prompt distribution computed from each of the three models: SD-1.4 (Rombach et al., 2022), SDXL-Turbo (Sauer et al., 2023), and FLUX.1-dev. We take the majority decision for each example and find that GPT-4o aligns with the majority decision **92.8%** of the time: SD-1.4 88% of the time, FLUX.1-dev (Labs, 2024) 91.2%, and SDXL-Turbo 99.5%. In both of our human evaluation experiments, GPT-4o agrees with human evaluation in over 90% of the cases, across multiple images. This confirms that it is a reliable underlying VQA model.

We provide further details on human evaluation in Appendix G.

## 4.1 COMPARING GRADE TO PREVIOUS METRICS

After establishing the validity of GRADE, we now turn to compare it to FID and Recall and find that they are weakly correlated, which coupled with the high accuracy of GRADE, shows FID and Recall measures do not accurately capture semantic-level diversity.

To facilitate an apples-to-apples comparison, we modify GRADE to use references. By doing so, the score relies on the same reference set the baseline metrics use to compute their diversity score. We thus swap entropy in favor of Total Variation Distance (TVD). This modification does not fundamentally change GRADE, as the underlying estimated distributions are the same. We use LAION data

Table 2: **PCC between GRADE and traditional metrics, using CLIP.** FID has zero or low correlation with $TVD_G$, while Recall (R) exhibits negative correlation. This indicates the distributions estimated by GRADE capture a notion of diversity that existing metrics do not.

| Model | Dataset | FID-TVD$_G$ | R-TVD$_G$ |
|-------|---------|-------------|-----------|
| SD-1.1 | LAION-2B | 0.12 | -0.15 |
| SD-1.4 | LAION-2B | 0 | -0.20 |
| SD-2.1 | LAION-5B | 0 | -0.19 |

(Schuhmann et al., 2022) as a reference set and Pearson Correlation Coefficient (PCC) to determine whether GRADE correlates with FID and Recall.

Table 2 shows weak correlation with GRADE, which implies that FID and Recall do not accurately capture semantic-level diversity. Further detail, including experiments with Inception v3 (Szegedy et al., 2014), corroborating our results are presented in Appendix C.

## 5 COMPARING DIVERSITY OF TEXT-TO-IMAGE MODELS

We use GRADE to estimate the diversity of popular T2I models. We begin with an overview of our setup and then we present the results.

**Data and distributions overview.** For each model, we estimate distributions over 100 common concepts such as "cookie" and "suitcase" and attributes such as "shape" and "color". Each concept is linked to four questions on average. In total, there are 405 multi-prompt distributions and 2,430 single prompt distributions, consisting a total of 60,000 images per model.

**T2I models.** We use 12 models from three different families shown in Table 3. All models were used with their default hyperparameters as in the `Diffusers` library (von Platen et al., 2023).

Table 3: The 12 T2I Models grouped by family.

| Family | Model |
|--------|-------|
| **IF-DeepFloyd** | DeepFloyd-M, DeepFloyd-L, DeepFloyd-XL (at StabilityAI, 2023) |
| **Stable Diffusion** | SD-1.1, SD-1.4, SD-2.1 (Rombach et al., 2022), SDXL (Podell et al., 2023), SDXL-Turbo (Sauer et al., 2023), SDXL-LCM (Luo et al., 2023), SD-3 (2B) (Esser et al., 2024) |
| **FLUX** | FLUX.1-schnell (Black Forest Labs, 2024b), FLUX.1-dev (Black Forest Labs, 2024a) |

### 5.1 RESULTS

**All models have low diversity scores.** Table 4 presents the mean entropy of models across both multi- and single-prompt distributions, with with permutation tests showing the results are statistically significant in Appendix D.2. On average, multi-prompt distributions across all models have a mean entropy of 0.57, while single prompt distributions exhibit lower diversity with a mean entropy of approximately 0.44. Fig. 3 illustrates the differences in diversity between models, and additional examples are provided in Appendix A.

**Relation of diversity to model size.** The relationship between model size and diversity suggests that diversity decreases as model size increases, as illustrated in Fig. 4a. This trend indicates an *inverse-scaling law* (McKenzie et al., 2023), supported by Pearson $r = -0.7$ ($p = 0.011$) and Spearman $\rho = -0.84$ ($p = 0.001$) correlations between diversity and model size. However, given the small sample size of 12 models, and potential confounding factors, such as different data and architectures, we do not make any causal claims and these findings should be interpreted with caution. Furthermore, in addition to our claims in Section 6 (that underspecified captions cause low diversity), Fig. 4b shows that the more a model generates images that cannot be answered[1] (i.e., prompt adherence *decreases*), the more diverse it is. Pearson $r = 0.8$ ($p = 0.02$) and Spearman $\rho = 0.94$ ($p < 0.001$) correlations reinforce this, suggesting the possibility that improving the ability of models to generate images that match the prompt is at the cost of sample diversity, similar to fidelity-diversity tradeoffs shown before (Dhariwal & Nichol, 2021; Kynkäänniemi et al., 2019).

---

[1]In Section 4 we show that 80% of unanswerable images do not depict the concept mentioned in the prompt.

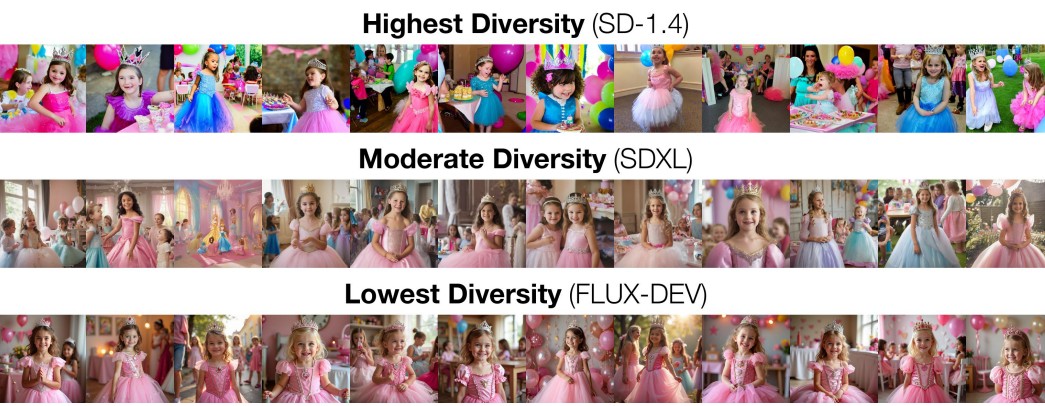

Figure 3: **Difference in diversity between models.** Images generated using the prompt "a princess at a children's party". Each row corresponds to a model, top-down: SD-1.4 (most diverse), SDXL, and FLUX.1-dev (least diverse). While no model exhibits high diversity, there is a marked difference between SD-1.4 and FLUX.1-dev, with SDXL between them. Specifically, diversity is reduced in attributes such as the ethnicities of depicted people, colors of dresses, and overall backgrounds.

Table 4: **Entropy in multi- and single-prompt distributions.** The mean entropy, standard deviation, and standard error of the mean (SEM) over all distributions for each model over multi-prompt and single-prompt distributions. Values are presented as mean ± SD (SEM). Values close to 1 indicate highly diverse behavior (uniform) while values close to 0 indicate highly repetitive categories. The *most* diverse models are in bold.

| | Mean Entropy ↑ | |
| --- | --- | --- |
| **Model** | **Multi-prompt** | **Single-prompt** |
| DeepFloyd-M | **0.64 ± 0.30 (0.01)** | 0.49 ± 0.34 (0.00) |
| DeepFloyd-L | 0.62 ± 0.29 (0.01) | 0.47 ± 0.34 (0.00) |
| DeepFloyd-XL | 0.61 ± 0.30 (0.01) | 0.46 ± 0.34 (0.00) |
| SD-1.1 | **0.64 ± 0.30 (0.01)** | **0.54 ± 0.33 (0.00)** |
| SD-1.4 | **0.64 ± 0.29 (0.01)** | 0.53 ± 0.33 (0.00) |
| SD-2.1 | 0.63 ± 0.30 (0.02) | 0.51 ± 0.34 (0.00) |
| SDXL | 0.59 ± 0.31 (0.02) | 0.46 ± 0.34 (0.00) |
| SDXL-Turbo | 0.52 ± 0.33 (0.02) | 0.36 ± 0.33 (0.00) |
| SDXL-LCM | 0.58 ± 0.32 (0.02) | 0.45 ± 0.34 (0.00) |
| SD-3 (2B) | 0.47 ± 0.33 (0.02) | 0.34 ± 0.33 (0.00) |
| FLUX.1-schnell | 0.48 ± 0.33 (0.02) | 0.36 ± 0.33 (0.00) |
| FLUX.1-dev | 0.47 ± 0.33 (0.02) | 0.32 ± 0.32 (0.00) |

**Default behaviors.** We define *default behavior* as a phenomenon where a model has a heavily skewed distribution toward a specific attribute $\tau \geq 80\%$ of the time. We observe that default behaviors are highly frequent and maintain the trends in Fig. 4, indicating strong correlation to entropy. All models exhibit at least one default behavior from **76% to 90%** of the multi-level distributions and from **87% to 97%** of the single prompt distributions. Similarly, the range of total default behaviors exhibited in multi-prompt distributions is between **39% to 56%** and between **49% to 70%** for single prompt distributions. Complete results with further analyses are provided in Appendix E.

## 6 IS LOW DIVERSITY ROOTED IN THE TRAINING DATA?

In the previous section we showed T2I models exhibit low diversity. We believe the origin of this behavior is from the training data: images for a given concept where the prompt does not specify an attribute value, are not diverse with respect to its attribute values. Specifically, each example in the training set of a T2I model consists of a caption-image pair. Captions mentioning a concept can leave it unspecified ("banana") or specify an attribute value ("yellow banana"). We assume the

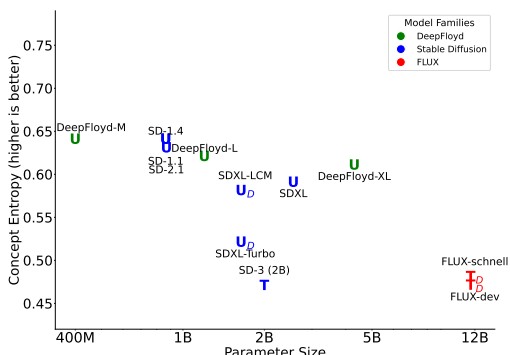 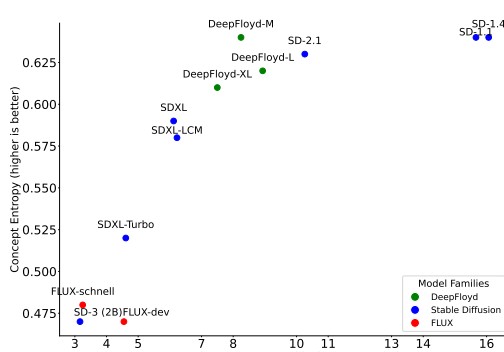

(a) The mean concept entropy of the models plotted against the denoiser's parameter size. To a degree, diversity deteriorates in tandem with parameter size. This effect is most apparent within every model family. Models marked with $U$ denote U-Net-based models, $T$ denotes transformer-based models. $U_D$ and $T_D$ denote distilled models.

(b) Mean concept entropy plotted against the % of answers mapped to "none of the above" (in Section 4 we show 80% of which account for missing concepts in the image). Low "none of the above" values correspond to *high* prompt adherence. The plot illustrates the relationship between diversity and prompt adherence, suggesting a tradeoff between the two.

Figure 4: Comparison of mean concept entropy across different models: (a) relative to the denoiser's parameter size and (b) in relation to prompt adherence.

model learns a different text-image distribution for each case. Since our prompts are underspecified, we focus on the first kind. We hypothesize that the training image distributions in such cases (where the attribute is unspecified) are relatively not diverse, and that the model learns to replicate this behavior. Indeed, we see this anecdotally in 100 samples in LAION. This behavior can be attributed to the linguistic phenomenon of *reporting bias* (Gordon & Van Durme, 2013).

To test this hypothesis, we use GRADE to measure diversity of multi-prompt distributions of training data images in LAION, for images whose captions mention the concept without specifying an attribute. We then compare these distributions with the distribution of generated images from a given model, also computed with GRADE.

Our analysis consists of two steps: first, we test if the models replicate the multi-prompt distributions based on *underspecified caption-image pairs in the training data*, by comparing them to multi-prompt distributions based on *generated images with the same captions*; second, we test if this replication extends to images generated with *underspecified prompts not found in the training set*.

**Replicating training data diversity.** We aim to assess whether models reproduce the underspecified diversity of their training data. For each model, we sample 50 triplets of concepts (e.g., "cookie"), attributes (e.g., "shape"), and attribute values, and measure their diversity distribution using GRADE. Then, using the same concepts and attributes we sample captions from the corresponding LAION dataset that explicitly mention the concept but not the attribute. To ensure the captions meet our criteria, we apply two filtering conditions: (1) the concept must be mentioned as an object and not as a modifier (e.g., "cookie" but not "cookie cutter"), and (2) the caption must not mention or imply the attribute of interest (e.g., "a classic chocolate chip cookie" implies the cookie is round). After filtering, we remain with 150 captions per concept and compute their distributions (full details are presented in Appendix F). We generate 20 images per caption, resulting in 3,000 images per concept and then estimate their distribution. Finally, we compute the TVD between these distributions and evaluate how closely the models' outputs mirror the diversity present in their corresponding, underspecified training data.

**Generalizing to underspecified prompts.** We examine whether the model's ability to replicate the diversity in the training data extends to new, similarly underspecified prompts. To achieve this, we compare the multi-prompt distributions from LAION images with those derived from prompts and images generated by GRADE, each consisting of 600 images, as discussed in Section 5. These prompts, like the filtered captions, mention the concept without specifying the attribute (e.g., "a cookie in a bakery"). By calculating the TVD between these distributions, we assess whether models have generalize the learned concept-attribute associations to new, underspecified prompts.

Table 5: **Similarities between model outputs and its training set.** The entropy values, PCC, and TVD all indicate models have comparable diversity to the training set.

| Model | Dataset | Source of Prompts | Model Entropy | Dataset Entropy | PCC | TVD |
|-------|---------|-------------------|---------------|-----------------|------|------|
| SD-1.1 | LAION-2B | LAION-2B | 0.62 | 0.64 | 0.86 | 0.11 |
|        |          | Generated | 0.58 |      | 0.71 | 0.18 |
| SD-1.4 | LAION-2B | LAION-2B | 0.62 | 0.64 | 0.88 | 0.10 |
|        |          | Generated | 0.60 |      | 0.72 | 0.17 |
| SD-2.1 | LAION-5B | LAION-5B | 0.68 | 0.65 | 0.73 | 0.13 |
|        |          | Generated | 0.68 |      | 0.61 | 0.18 |

## 6.1 RESULTS

The diversity in the generated images closely corresponds to the diversity in the training data (see Table 5). LAION exhibits moderate diversity, with dataset entropy values of 0.64 for LAION-2B and 0.65 for LAION-5B. The models' entropy values when using LAION captions as prompts are comparable, ranging from 0.62 to 0.68. High PCC, ranging from 0.73 to 0.88, indicate a strong correlation between the multi-prompt distributions of the generated images and those in the training data. The low TVD values (0.10 to 0.13) further suggest that the distributions are similar.

When using the generated prompts instead of LAION captions, we observe a slight decrease in PCC (ranging from 0.61 to 0.72) and a modest increase in TVD (ranging from 0.17 to 0.18), but the overall trend remains consistent. This indicates that the models continue to reflect the training data's multi-prompt distributions even when presented with new, underspecified prompts.

These findings support our hypothesis that the low diversity observed in the generated images is rooted in the underspecified training captions. Models learn to associate concepts with their most common attribute values since those are unspecified in captions but are present in the corresponding images, leading to a lack of diversity in the outputs that models learn to mimic.

## 7 LIMITATIONS

There are three main limitations to GRADE. First, because GRADE considers specific attributes of specific concepts, the diversity scores do not reflect the diversity of concepts and attributes that were not measured. Second, GRADE relies on an underlying LLM and VQA, which have unknown biases that influence the suggestions by the LLM and affect the quality of information extracted by the VQA. Finally, while our definition of diversity is clearly defined, human perception of diversity is an open question, and we do not know if it aligns with GRADE.

## 8 CONCLUSION

We introduce GRADE, an automated fine-grained evaluation method for measuring sample diversity in T2I models based on concepts and their attributes. By estimating the distribution of attributes in generated images for a given concept and computing entropy, GRADE provides a diversity score that can be used to interpret model behavior. Unlike traditional diversity metrics, GRADE does not rely on reference images. Our experiments demonstrate that humans GRADE is accurate, while at the same time showing weak correlation with traditional metrics like FID and Recall. We conduct a comprehensive analysis of 12 state-of-the-art T2I models and uncovered a prevalent limitation: these models default to generating images with the same attributes for a concept on anywhere from 78% to 90% of the concepts we tested, with an increasing trend as models scale in size and improve in prompt adherence, highlighting a limited ability to capture the rich diversity inherent in visual concepts. We further hypothesize that diversity in generation is linked to diversity in the training data, and that under-specification encourages default behavior. Future work could explore methods to enrich training data, incorporate diversity-promoting mechanisms during model training, and extend GRADE to evaluate relationships between different attributes of a concept or the relationship between multiple concepts in a scene. We hope our work will inspire more nuanced evaluations and drive advancements in generating diverse visual content from textual descriptions.

ETHICS STATEMENT

This research evaluates the diversity of text-to-image (T2I) models using our proposed method, GRADE. We note that T2I models may inadvertently perpetuate biases present in their training data, leading to less diverse or stereotypical outputs. By quantifying sample diversity and identifying default behaviors, our work aims to promote fairness and inclusivity in generative modeling. At the same time, the pipeline we propose does not capture all human notions of diversity, and models should not be judged as "diverse enough" based on it alone in application where output diversity is crucial.

All datasets and models used are open-source. Human evaluations were conducted with informed consent, and participants were fairly compensated (an hourly rate of $15).

REPRODUCIBILITY STATEMENT

All necessary information to reproduce GRADE is provided in Section 3, and the complete set of concepts, prompts, attributes, and attribute values are included in our submission. The code and data for this paper will be open-sourced.

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

# A QUALITATIVE EXAMPLES OF DIVERSITY

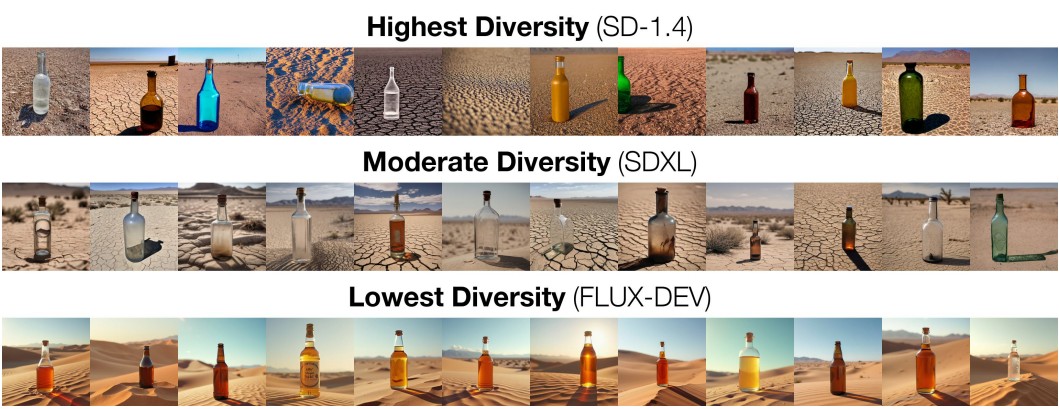

Figure 5: **Difference in diversity between models.** Images generated using the prompt "a bag on a cliffside". Each row corresponds to a model, top-down: SD-1.4 (most diverse), SDXL, and FLUX.1-dev (least diverse). While no model exhibits high diversity, there is a marked difference between SD-1.4 and FLUX.1-dev, with SDXL between them. Specifically, diversity is reduced in attributes such as color and placement of the bags, as well as the background.

Figure 6: **Difference in diversity between models.** Images generated using the prompt "a bottle in a desert". Each row corresponds to a model, top-down: SD-1.4 (most diverse), SDXL, and FLUX.1-dev (least diverse). While no model exhibits high diversity, there is a marked difference between SD-1.4 and FLUX.1-dev, with SDXL between them. Here, the lack of diversity is most pronounced in the color of the bottle or its liquid. While SD-1.4 depicts relatively varied bottles, SDXL depicts transparent ones, while FLUX.1-dev depicts almost exclusively orange-like bottles.

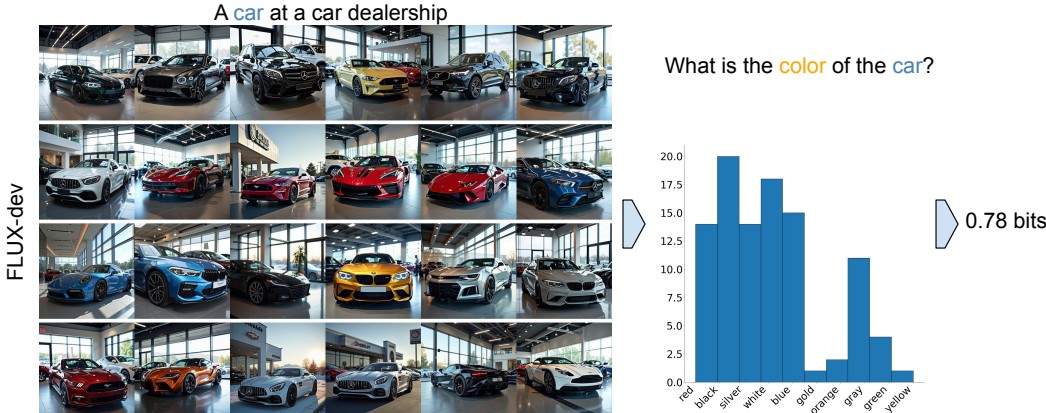

Figure 7: **Illustration of diversity score.** Displayed are 24 of the 100 images generated by FLUX.1-dev using the prompt "A car in a car dealership". The accompanying histogram and the subsequent entropy plot both represent the 100. The diversity score is 0.78, indicating the color of the cars is relatively diverse.

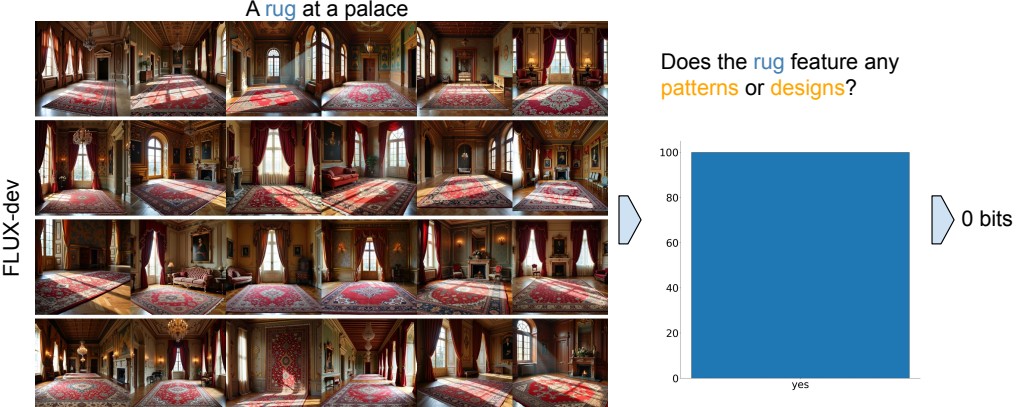

Figure 8: **Illustration of diversity score.** Displayed are 24 of the 100 images generated by FLUX.1-dev using the prompt "A rug at a palace". The accompanying histogram and the subsequent entropy plot both represent the 100. The diversity score is 0, indicating the rugs are consistently patterned.

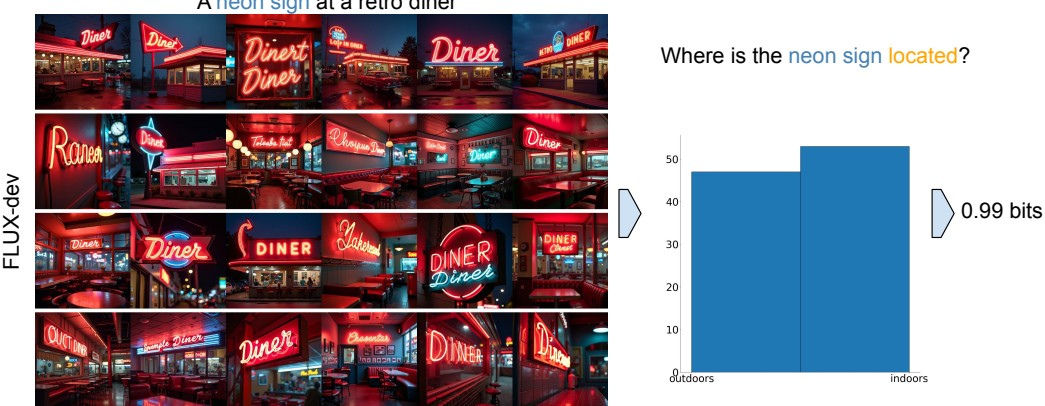

Figure 9: **Illustration of diversity score.** Displayed are 24 of the 100 images generated by FLUX.1-dev using the prompt "A neon sign at a retro diner". The accompanying histogram and the subsequent entropy plot both represent the 100. The diversity score is 0.99, indicating the location of the signs is uniform.

# B    EXTENDED DATA OVERVIEW

Table 6: Concepts and their attributes with detailed distributions.

| Concept | Question (Attribute) | Attribute Values |
|---|---|---|
| Bin | What shape is the bin? | circular, octagonal, square, cylindrical, triangular, rectangular, round, oval, hexagonal |
| | Does the bin have a lid? | yes, no |
| Person | Is the person male or female? | male, female |
| | Does the image show the person from up-close? | yes, no |
| Suitcase | Is the suitcase open or closed? | open, closed |
| | Is the suitcase soft-shell or hard-shell? | soft-shell suitcase, hard-shell suitcase |
| Cake | Does the cake have multiple tiers? | yes, no |
| | Is the cake eaten? | |
| Pool | Is there anyone swimming in the pool? | yes, no |
| | What color is the water in the pool? | reflective like a mirror, black, clear, green, blue, brown |

# C  COMPARING GRADE TO PREVIOUS METRICS

Table 7: **PCC between GRADE and traditional metrics, using CLIP.** FID, Recall (R), and Precision (P) show low to moderate degrees of correlation among each other, while the TVD based on the distributions from GRADE (TVD$_G$) exhibits weak correlations with all of them. This indicates the distributions estimated by GRADE capture diversity existing metrics do not.

| Model | Dataset | FID-R | FID-P | R-P | FID-TVD$_G$ | R-TVD$_G$ | P-TVD$_G$ |
|---|---|---|---|---|---|---|---|
| SD-1.1 | LAION-2B | 0.14 | -0.15 | 0 | 0.12 | -0.15 | 0 |
| SD-1.4 | LAION-2B | 0.19 | -0.40 | 0 | 0 | -0.20 | -0.15 |
| SD-2.1 | LAION-5B | -0.21 | -0.48 | 0 | 0 | -0.19 | 0.15 |

Table 8: **Evaluation results with existing metrics using CLIP.** Each value in the table is the mean of the metric over the 50 pairs of multi-prompt distributions.

| Model | Dataset | TVD$_G$ | FID | Recall | Precision |
|---|---|---|---|---|---|
| SD-1.1 | LAION-2B | 0.15 | 290 | 0.12 | 0.88 |
| SD-1.4 | LAION-2B | 0.15 | 276 | 0.15 | 0.92 |
| SD-2.1 | LAION-5B | 0.16 | 290 | 0.12 | 0.94 |

Table 9: **PCC between GRADE and traditional metrics, using Inception v3.** FID, Recall (R), and Precision (P) show low to moderate degrees of correlation among each other, while the TVD based on the distributions from GRADE (TVD$_G$) exhibits weak correlations with all of them. This indicates the distributions estimated by GRADE capture diversity existing metrics do not.

| Model | Dataset | FID-R | FID-P | R-P | FID-TVD$_G$ | R-TVD$_G$ | P-TVD$_G$ |
|---|---|---|---|---|---|---|---|
| SD-1.1 | LAION-2B | -0.41 | 0.23 | -0.34 | 0.14 | 0.04 | 0 |
| SD-1.4 | LAION-2B | -0.48 | 0.14 | -0.22 | 0.18 | -0.10 | 0.14 |
| SD-2.1 | LAION-5B | -0.12 | -0.52 | 0 | -0.16 | -0.15 | 0.13 |

Extended results for the comparison between GRADE and traditional metrics described in Section 4.1. Results using CLIP for feature extraction can be viewed in Table 7 and Table 8. Results using Inception v3 (Szegedy et al., 2014) (ImageNet features (Deng et al., 2009)) are in Table 9 and Table 10. Below we detail the process of collecting the image sets and comparing between them.

**Reference and generated images.**  Since LAION is opensource and was used to train SD-1.1, SD-1.4, and SD-2.1; LAION-2B for the first two and LAION-5B for the latter–we sample images from it and compare them to images generated by the models. Specifically, we sample 50 from the 405 multi-prompt distributions (i.e., only the concept, attribute, and attribute values, not the prompts and images) in Section 5. Next, we sample 115 image and caption pairs using WIMBD, where the image depicts the concept and the caption mentions the concept but not the attribute, in accordance with our approach (Section 3.1). We end up with 50 reference distributions, each consisting of 115 images. To get our generated images, we generate one image for each caption, to maintain equal proportion between the distributions. For example, if an image in LAION is linked to the caption "Unicorn Cookie", its corresponding distribution will contain an image that was generated using that caption as a prompt.

**Details of metrics.**  Using the 50 pairs of distributions, we can compare GRADE to the metrics. Since entropy is not a reference-based metric, we change it in favor of Total Variation Distance (TVD) and use it on top of the distributions estimated by GRADE. We compute FID and Recall, using features from the open-clip implementation (the `ViT-H/14` variant) (Ilharco et al., 2021; Radford et al., 2021), trained on LAION-2B. Recall was computed with $k = 3$. We run the same experiment using Inception v3 features with $64$ dimensions.

Table 10: **Evaluation results with existing metrics using Inception v3.** Each value in the table is the mean of the metric over the 50 pairs of distributions.

| Model | Dataset | $\text{TVD}_G$ | FID | Recall | Precision |
|-------|---------|------|-----|--------|-----------|
| SD-1.1 | LAION-2B | 0.15 | 19.67 | 0.35 | 0.75 |
| SD-1.4 | LAION-2B | 0.15 | 15.0 | 0.45 | 0.74 |
| SD-2.1 | LAION-5B | 0.16 | 18.67 | 0.49 | 0.83 |

## C.1 QUALITATIVE METRIC COMPARISON EXAMPLES

Is the picnic basket made of wicker?  $\text{TVD}_\text{GRADE}$ **= 0**    FID = 272    Recall = 0    Precision = 0.9

LAION-2B

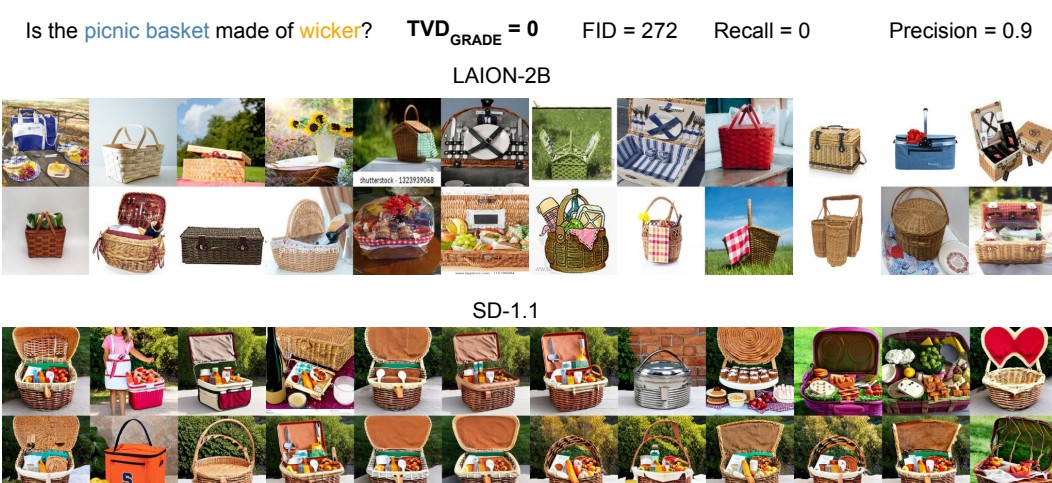

SD-1.1

Figure 10: **Comparison between GRADE, FID, and Recall, using CLIP features.** The metrics are compared over the "wicker" attribute of the concept "picnic basket". $\text{TVD}_\text{GRADE}$ reports very high similarity between the sets of images, which is indeed shown in the images (almost all picnic baskets are made of wicker). In contrast, Recall and FID report very low scores.

Are there any visible stains or damage on the tablecloth?  $\text{TVD}_\text{GRADE}$ **= 0.03**  FID = 240    Recall = 0.08    Precision = 0.93

LAION-2B

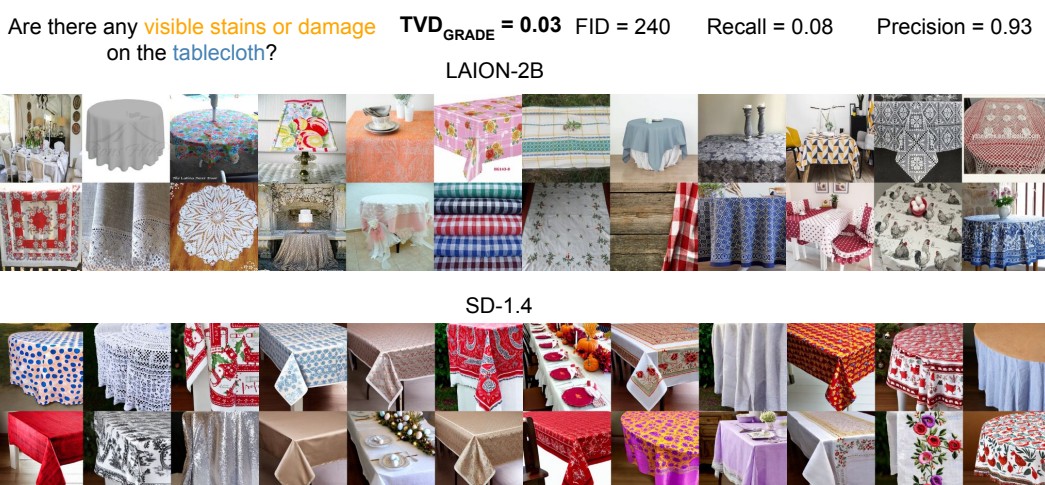

SD-1.4

Figure 11: **Comparison between GRADE, FID, and Recall, using CLIP features.** The metrics are compared over the "visible stains or damage" attribute of the "tablecloth" concept. $\text{TVD}_\text{GRADE}$ reports very high similarity between the sets of images, which is indeed shown in the images (the tablecloth is rarely damaged in either set). In contrast, Recall and FID report very low scores.

# D EXTENDED RESULTS

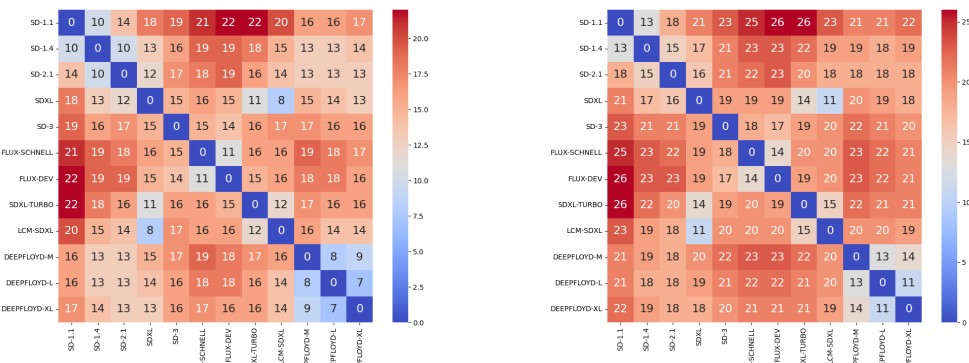

(a) The mean TVD between all pairs of models over multi-prompt distributions.

(b) The mean TVD between all pairs of models over single prompt distributions.

Figure 12: For readability, both figures show TVD in a range between 0 and 100 instead of 0 to 1.

Table 11: **Backbones, their associated models, and the average mean TVD of models with a shared backbone.**

| Backbone | Models | Average mean TVD |
|---|---|---|
| SD-1.1 | SD-1.1, SD-1.4, SD-2.1 | 11 |
| SDXL | SDXL, SDXL-LCM, SDXL | 10 |
| FLUX | FLUX.1-schnell, FLUX.1-dev | 11 |
| DeepFloyd | DeepFloyd-M, DeepFloyd-L, DeepFloyd-XL | 8 |

**Similarity in diversity across distributions.** We investigate the similarity in diversity across models we find in Section 5.1. We modify GRADE to use Total Variation Distance (TVD) instead of entropy to facilitate comparisons between corresponding distributions in the attribute value level. For example, the difference between the frequency of "blue" in the multi-prompt distribution of the concept *tie* and attribute *color*. Results for both concept and single prompt distributions are shown in Fig. 12. The results are in line with our other findings: all models have similar distributions, with the maximum TVD for multi-prompt distributions being 0.22 and for single prompt distributions 0.26, with these numbers being the result of a comparison between the least and most diverse models (i.e., SD-1.1 and FLUX.1-dev). Moreover, models with similar backbone have smaller distances. The groups and the mean TVDs are shown in Table 11.

## D.1 ADDITIONAL ANALYSIS ON MODEL SIZE

We further investigate the relationship between model size and diversity, and prompt adherence and diversity. Fig. 13 shows that as the denoisers' parameter size increases, both the mean concept entropy and the mean prompt entropy decrease. This suggests that larger models produce less diverse outputs, indicating an inverse-scaling law (McKenzie et al., 2023). The negative correlation is supported by significant Pearson and Spearman correlation coefficients at both the concept level (Pearson $r = -0.701$, $p = 0.011$; Spearman $\rho = -0.842$, $p = 0.001$) and the prompt level (Pearson $r = -0.666$, $p = 0.018$; Spearman $\rho = -0.804$, $p = 0.002$).

Figure 14 illustrates negative correlation between diversity and prompt adherence. As the percentage of unanswerable images ("none of the above") increases i.e., prompt adherence *decreases*, the diversity measured by entropy increases. This is quantified by strong positive Pearson and Spearman correlations at both the concept level (Pearson $r = 0.802$, $p = 0.002$; Spearman $\rho = 0.938$, ($p < 0.001$) and the prompt level (Pearson $r = 0.871$, ($p < 0.001$); Spearman $\rho = 0.947$, ($p < 0.001$). This

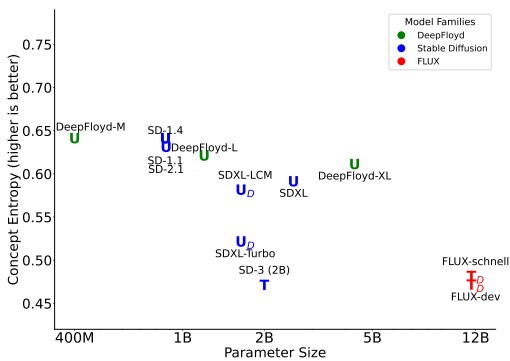 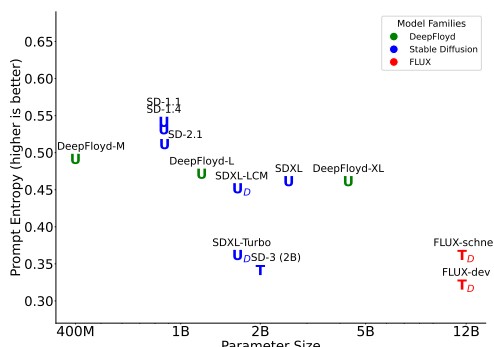

(a) The mean concept entropy of the models plotted against the denoiser's parameter size.

(b) The mean prompt entropy of the models plotted against the denoiser's parameter size.

Figure 13: (a) The mean concept entropy of the models plotted against the denoiser's parameter size. (b) The mean prompt entropy of the models plotted against the denoiser's parameter size. To a degree, diversity deteriorates in tandem with parameter size. This phenomenon is most apparent within every model family. Models marked with $U$ denote U-Net-based models, $T$ denotes transformer-based models. $U_D$ and $T_D$ denote distilled models.

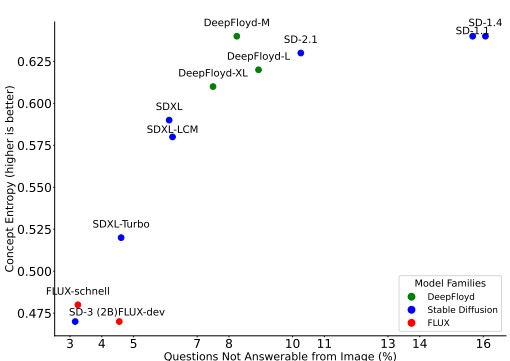 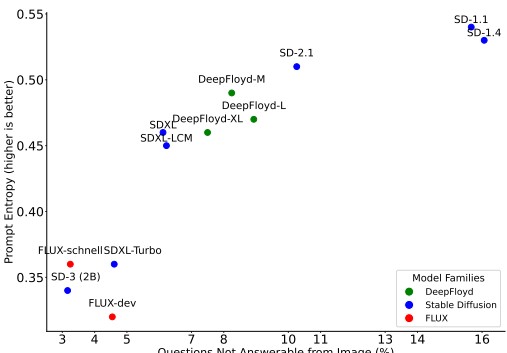

(a) The mean concept entropy of the models plotted against the % of "none of the above".

(b) The mean prompt entropy of the models plotted against the % of "none of the above".

Figure 14: (a) The mean concept entropy of the models plotted against the % of "none of the above". (b) The mean prompt entropy of the models plotted against the % of "none of the above". In Section 4 we show 80% of which account for missing concepts in the image. The plots show negative correlation between diversity and prompt adherence, which indicates there is a tradeoff.

indicates a trade-off between diversity and prompt adherence: models that generate more diverse outputs tend to adhere less strictly to the prompts.

## D.2 STATISTICAL SIGNIFICANCE OF DIVERSITY SCORES

To confirm that our results are statistically significant, we perform a two-tailed permutation test between every unique pair of models for both distribution types (single-prompt and multi-prompt). This test is common when the data comes from a complex distribution (Bonnini et al., 2024), in our case, the distribution of diversity scores of each model. We demonstrate that the difference between the vast majority of models is statistically significant in both cases.

Concretely, there are 66 unique model pairs. For each pair, we compute a two-tailed permutation test with the null hypothesis $H_0$ that the diversity scores of the two models are the same. We perform

$N = 100{,}000$ permutations, where the p-value is defined as

$$p = \frac{\text{number of permutations where } |D_{\text{perm}}| \geq |D_{\text{obs}}|}{N},$$

where $D_{\text{obs}}$ is the observed difference in diversity scores between the two models, and $D_{\text{perm}}$ is the difference obtained under each permutation. We compare the p-value $p$ to a significance level of $\alpha = 0.05$.

**Results.** The vast majority of pairs are statistically significant.

Comparisons based on single-prompt distributions reveal just three pairs are not statistically significant: (SDXL, SDXL-LCM), (SDXL, DeepFloyd-XL), and (SDXL-Turbo, FLUX-schnell).

Similarly, comparisons using multi-prompt distributions, reveal only 15 pairs are not statistically significant. Non-significant pairs are similar in quality. For example, all pair combinations of SD-1.1, SD-1.4, and SD-2.1 are not significant, which is not surprising since these models largely share the same underlying architectures and training data.

### D.3 DISCUSSION OF RESULTS

Our findings reinforce the observations made in the main text regarding the interplay between model scale, diversity, and prompt adherence:

**Inverse-scaling law.** There is a negative correlation between diversity and model size, suggesting that increasing model parameters leads to decreased diversity. This phenomenon is most apparent within each model family and aligns with the concept of an inverse-scaling law.

**Fidelity-diversity trade-off.** The negative correlation between diversity and prompt adherence indicates a trade-off between a model's ability to generate images that match the prompt and the diversity of its outputs. This is consistent with previous findings on fidelity-diversity trade-offs (Dhariwal & Nichol, 2021; Kynkäänniemi et al., 2019), where improving a model's prompt-specific generation may reduce the overall diversity of its outputs.

## E   DEFAULT BEHAVIORS: EXTENDED RESULTS

In Section 5.1 we define default behaviors and mention that almost all concepts are associated with at least one default behavior, as shown in Table 12. In Table 13, we report the total number of default behaviors for both types of distributions.

Table 14 shows a sample of default behaviors detected in multi-prompt distributions and Fig. 15 images of these behaviors.

Table 12: **Percentage of at least one default behavior.** Lower values indicate higher diversity. Almost all concepts are associated with at least one default behavior in single prompt distributions, with a similar trend in multi-prompt distributions. The model with the *most* default behaviors is in bold. Results are rounded to the closest integer.

| Model | % of Default Behavior ↓ | |
| --- | --- | --- |
| | **Multi-prompt** | **Single-prompt** |
| DeepFloyd-M | 83 | 92 |
| DeepFloyd-L | 81 | 92 |
| DeepFloyd-XL | 80 | 92 |
| SD-1.1 | 78 | 87 |
| SD-1.4 | 82 | 87 |
| SD-2.1 | 76 | 89 |
| SDXL | 81 | 90 |
| SDXL-Turbo | 86 | 95 |
| SDXL-LCM | 82 | 92 |
| SD-3 (2B) | 88 | 95 |
| FLUX.1-schnell | **90** | **97** |
| FLUX.1-dev | 88 | 96 |

Table 13: **Percentage of all default behaviors.** Lower values indicate higher diversity. There are 405 multi-prompt and 2430 single prompt distributions in total. The table quantifies the total percentage of default behaviors observed. The model with the *most* default behaviors is in bold. Results are rounded to the closest integer.

| Model | % of Default Behavior ↓ | |
| --- | --- | --- |
| | **Multi-prompt** | **Single-prompt** |
| DeepFloyd-M | 39 | 54 |
| DeepFloyd-L | 39 | 56 |
| DeepFloyd-XL | 40 | 56 |
| SD-1.1 | 39 | 49 |
| SD-1.4 | 40 | 51 |
| SD-2.1 | 40 | 52 |
| SDXL | 44 | 57 |
| SDXL-Turbo | 50 | 67 |
| SDXL-LCM | 44 | 57 |
| SD-3 (2B) | 56 | 69 |
| FLUX.1-schnell | 55 | 67 |
| FLUX.1-dev | **56** | **70** |

Table 14: **A random sample of default behaviors.** The concept is underlined in the question column. Images corresponding to the behaviors in the table can be viewed in Fig. 15.

| Model | Question (Attribute) | Attribute Value | Percentage |
| --- | --- | --- | --- |
| SD-1.1 | Is the brick alone or in a stack with others? | stacked | 97.4 |
| SD-1.4 | Is there a frame around the mirror? | yes | 92.9 |
| SD-2.1 | Is the suitcase soft-shell or hard-shell? | hard-shell | 88.3 |
| SDXL | Is the detective female or male? | male | 99.6 |
| SD-3 (2B) | Is the tie a necktie or a bowtie? | necktie | 100 |
| FLUX.1-schnell | Is the clock analog or digital? | analog | 100 |

1242
1243
1244
1245
1246
1247
1248
1249
1250
1251
1252
1253
1254
1255
1256
1257
1258
1259
1260
1261
1262
1263
1264
1265
1266
1267
1268
1269
1270
1271
1272
1273
1274
1275
1276

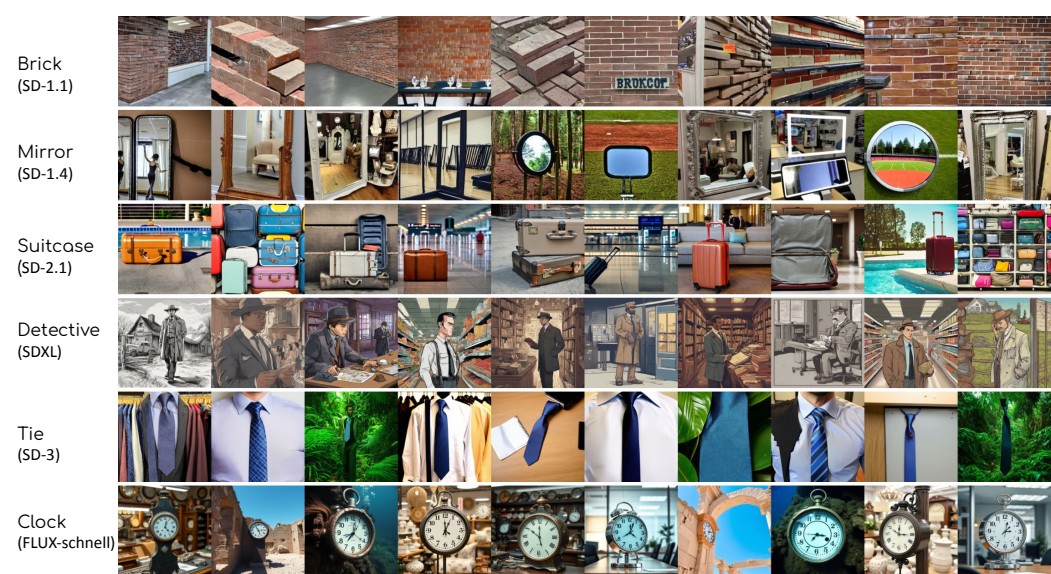

Figure 15: **A sample of images depicting the default behaviors in Table 14.** The concept is shown in the left column with the model directly below it. Images were sampled randomly from all prompts. The default behaviors, top down: (1) stacked bricks; (2) framed mirrors; (3) hard-shell suitcase; (3) male detective; (4) neckties; and (5) analog clocks.

1281
1282
1283
1284
1285
1286
1287
1288
1289
1290
1291
1292
1293
1294
1295

# F   IS LOW DIVERSITY ROOTED IN THE TRAINING DATA?

**Filtering Captions from LAION.**   We aimed to measure the diversity of training images whose captions satisfy two conditions: (1) they mention the concept as an object and not as a modifier (e.g., "cookie" but not "cookie cutter"), and (2) the caption must not mention or imply the attribute of interest (e.g., "a classic chocolate chip cookie" implies the cookie is round). We queried LAION using WIMBD (Elazar et al., 2024) and sampled 500 captions for each concept.

To efficiently filter the captions, we utilized GPT-4o in a few-shot setup. For each caption, we provided the caption text, the concept (e.g., "cookie"), and the question regarding the attribute of interest (e.g., "what is the shape of the cookie?"). We instructed GPT-4o to analyze each caption and determine whether it satisfies both filtering conditions. The model was prompted to reply with "yes" if both conditions are met and "no" otherwise.

We then downloaded the images associated with the captions that GPT-4o classified as satisfying both conditions. To ensure the reliability of our filtering method, we conducted a human evaluation, achieving an F1 score of 90.3%. Detailed methodology and results of the human evaluation are provided in Appendix G.

Below is the prompt we use with GPT-4o to filter captions from LAION:

```
In this task, you are provided with a caption associated with an
image, a concept, and a question. You need to find relevant captions
that do not indicate the answer to the question. Your role is
two-part. First, determine whether the caption explicitly mentions the
concept as a tangible thing, and not an accessory or an item related
to the concept. Second, determine if that question can be answered
only by reading the caption. If the answer is yes for the first and no
for the second, reply with "yes", otherwise reply with "no".

Here are some examples to guide your understanding:

Caption: teapot, glass teapot, Chinese teapot, herbal teapot, teaware
Concept: teapot
Question: What material is the teapot made of (ceramic, metal, glass,
etc.)?
Reasoning: The first part is to determine if teapot is mentioned in
the prompt. It is the first word in the caption, so it is. The second
part is to determine if the question is answerable from the prompt or
not. We want to find captions that are not answerable. Since there are
mentions of materials in the caption, it is answerable and the answer
is no.
Answer: no

Caption: My Sweet Angel Book Store Hyatt Book Store Amazon Books eBay
Book Book Store Book Fair Book Exhibition Sell your Book Book
Copyright Book Royalty Book ISBN Book Barcode How to Self Book
Concept: book
Question: Is the book dirty or clean?
Reasoning: The caption mentions items related to a book, but not an
actual book. The answer is no.
Answer: no

Caption: Perfect reading chair, cozy reading chair, nest chair, my
favorite chair, Nest Chair, Cozy Chair, Chair Cushions, Big Chair,
Cuddle Chair, Swivel Chair, Relax Chair, Big Comfy Chair, Chaise Chair
Concept: chair
Question: What color is the chair?
```

```
Reasoning: The first part is to identify if the caption mentions a
chair. It does mention a chair, with various adjectives. The second
part is to determine if the question is answerable from the caption.
The question asks about the color of the chair, and there is no
mention of a chair color. The answer is yes.
Answer: yes

Caption: JIX motorcycle helmet, cross helmet, full helmet, safety
helmet
Concept: helmet
Question: Does the helmet have any logos or graphics on it?
Reasoning: The first part is to determine if the caption mentions a
helmet. The caption indeed mentions a variety of helmets. The second
part is to determine if the question can be answered from the caption
alone. There is no information about logos or graphics in the caption,
so it is not answerable from the caption alone. The final answer is
yes because the answer to the first is yes and the second is no.
Answer: yes

Caption: dust bin, garbage container, recycle bin, trash icon
Concept: bin
Question: What shape is the bin?
Reasoning: The first part is to determine if the caption mentions a
bin. The caption mentions a bin, but it also mentions trash icon. This
indicates this is not an actual bin, but an icon of a bin. The answer
is no.
Answer: no

Caption: Cookie Policy – Cookie Law Compliance [MultiLang..
Concept: cookie
Question: What shape is the cookie?
Reasoning: The first part is to determine if the caption mentions a
cookie. The caption mentions cookie policy and cookie law compliance,
but not an actual edible cookie, that has a shape. The answer is no.
Answer: no

Caption: Best Cookie Presses – Cookie Press 150PCS Cookie Press Gun
with 16 Review
Concept: cookie
Question: Does the cookie have chocolate chips?
Reasoning: The first part is to determine if the caption mentions a
cookie or something else. The caption is about cookie press and not
actual cookie. The answer is no.
Answer: no
```

## G  HUMAN EVALUATION

**Worker selection.**   Workers were chosen based on their performance records, requiring them to have a minimum of 5,000 approved HITs and an approval rate above 98%. They had to achieve a perfect score on a qualification exam before being granted access to the task. An hourly wage of $15 was provided, ensuring they were fairly compensated for their efforts. In total, 71 unique workers participated in evaluating GRADE and 49 to filter the captions from LAION.

**Validating GRADE.**   To validate the VQA Section 4, we run an AMT crowdsourcing task where the worker is provided with a question, concept, image, and attribute values, and is requested to select the attribute value that best matches the question and image. The UI for this task can be viewed in Fig. 16 with examples in Fig. 17. A sample of cases from our attribute values coverage validation (validation of step (b)) is available in Fig. 18 and Fig. 19.

**Validating filtering of captions from LAION.** To assess the effectiveness of our GPT-4o-based caption filtering method described in Section 6, we conducted an Amazon Mechanical Turk (AMT) crowdsourcing task. We sampled 1,000 captions from LAION, ensuring an equal distribution of 500 captions that met the filtering criteria and 500 that did not. Workers were instructed to evaluate whether each caption (1) explicitly mentioned the concept as the main object rather than as a modifier (e.g., "cookie" instead of "cookie cutter") and (2) the caption must not mention or imply the attribute of interest (e.g., "a classic chocolate chip cookie" implies the cookie is round). Each example was reviewed by three independent workers, and the majority decision was taken as the final label. Our automated filtering method achieved a recall of 85.8% and a precision of 95.4%, resulting in an F1 score of 90.3%, which indicates a high level of agreement with human judgments. These findings demonstrate that GPT-4o is a reliable tool for automated caption filtering. Additional details about the user interface and example cases are provided in Fig. 20 and Fig. 21, respectively.

**Question:** *If there is ketchup or mustard, is it in wave form on the **hot dog**?*
**Options:** *'yes', 'no'*
**Correct Answer:** *yes*
**Explanation:** *The perspective may be confusing since we can't see the entire hot dog, but the mustard is laid out in what appears to be wave form. The answer is yes.*

**Main Task:**

Given the following image and question, **select the most appropriate answer** based on the image. If the image does not contain **${concept}** or none of the provided answer choices correctly describe the image, please select '**None of the above**'.

Main Task Image

**Question:** ${question}

**Options:**

| ${options} | ⌄ | **Submit** |

Figure 16: A screenshot of the VQA validation task. Workers are provided a question, concept, image, and a set of categories, including "none of the above" (options here). Their task is to select the option that answers the question.

**Instructions:** In this task, you will be provided with an image and a question. Your job is to select the correct answer to the question based on the options in the dropdown menu. If no option reasonably fits the question or the object you are asked about is not in the image, select the "None of the above" option. Below are examples of how to select an answer. Please use it as a guide for the main task that follows.

**Example 1:**

**Question:** *What type of **helmet** is depicted in the image (e.g., sports, construction, military)?*
**Options:** *['aviation helmets', 'diving helmets', 'motorcycle helmets', 'firefighter helmets', 'mining helmets', 'engineering helmets', 'construction helmets', 'ceremonial helmets', 'bicycle helmets', 'equestrian helmets', 'military helmets', 'skiing helmets', 'sports helmets']*
**Correct Answer:** *bicycle helmets*
**Explanation:** *The image shows a bicycle helmet.*

**Example 2:**

**Question:** *Is the **umbrella** open or closed?*
**Options:** *'closed', 'open'*
**Correct Answer:** *open*
**Explanation:** *The umbrella is open.*

**Example 3:**

**Question:** *Is the **drawer** open or closed?*
**Options:** *'open', 'closed'*
**Correct Answer:** *None of the above*
**Explanation:** *There is no drawer in the image, there is something that looks like a table, but it does not have an inner shelf for item storage.*

Figure 17: 3 out of 10 examples provided to workers as aid to complete their visual question answering task.

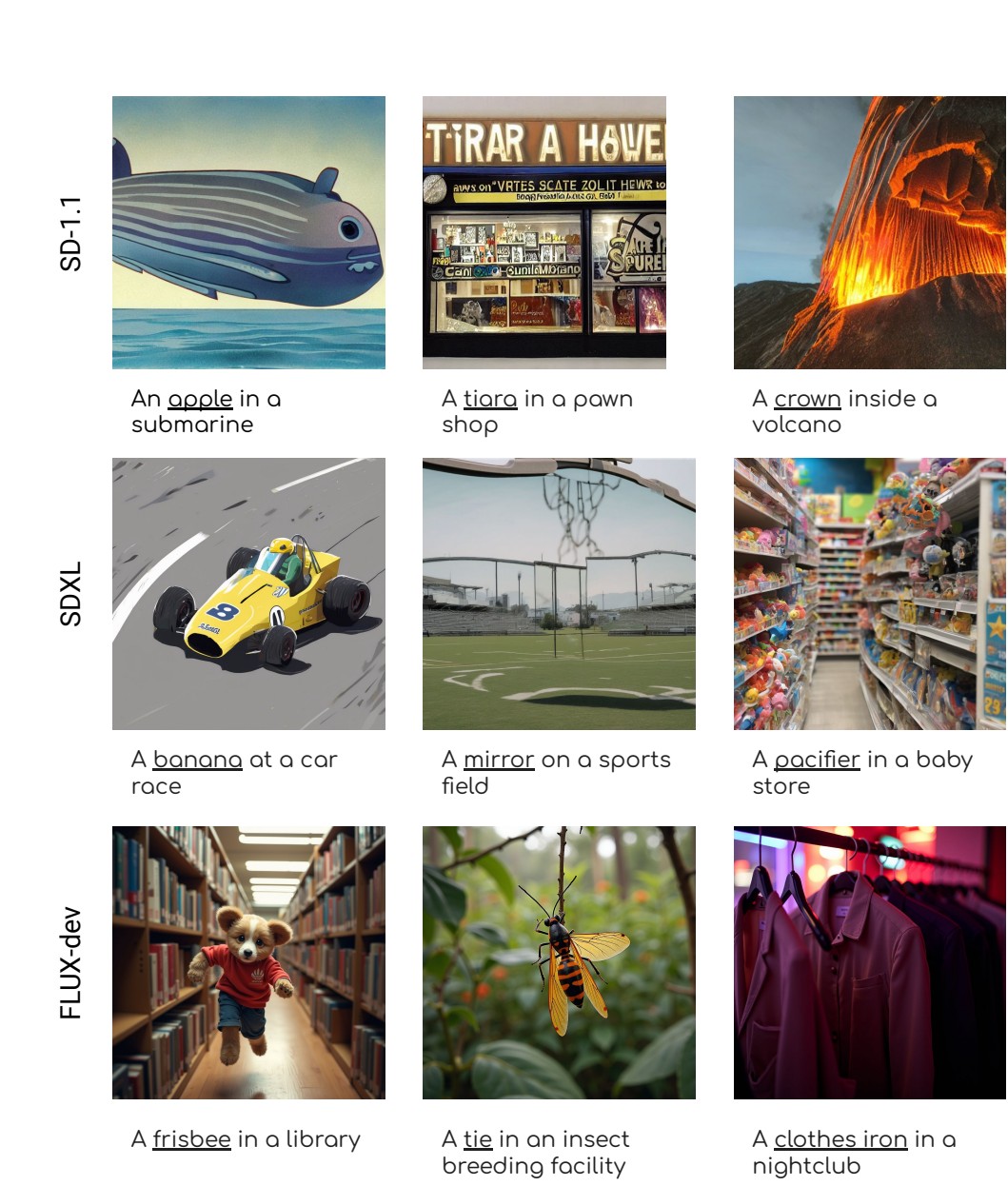

Figure 18: A sample of images marked with "none of the above", as a result of not including the concept (underlined) in the image.

SDXL

Popcorn at a cinema

Q: is the popcorn in a bowl or a bucket?

$V^a_c$ = {bucket, bowl}

SD-1.1

a toy at a children's playroom

Q: Does the toy appear to be mechanical or electronic?

$V^a_c$ = {mechanical, electronic}

SDXL

a tie in an office

Q: Is the tie worn with a formal or casual outfit?

$V^a_c$ = {casual, formal}

FLUX-dev

A person in a city square

Q: Is the person male or female?

$V^a_c$ = {male, female}

Figure 19: A sample of images marked with "none of the above". The top row exhibits cases where the attribute value is not in $\mathcal{V}^a_c$. The bottom row exhibits cases where the question cannot be answered just from viewing the image. The concept in each prompt is underlined.

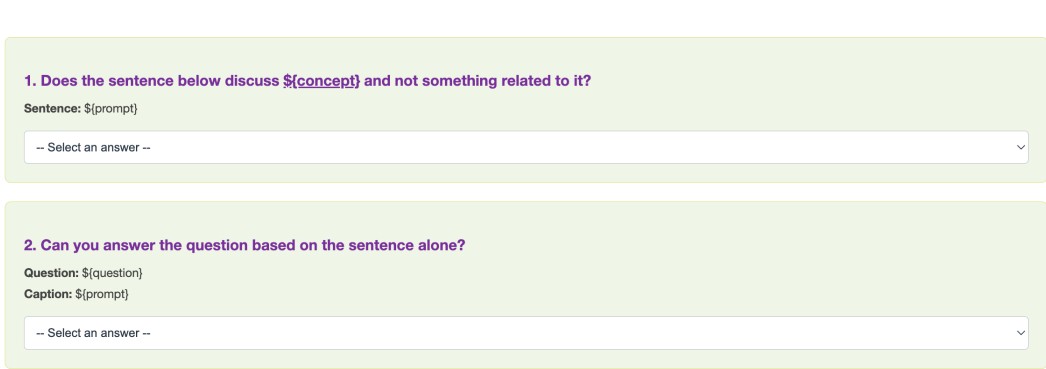

Figure 20: A screenshot of the caption filtering validation task. Workers are provided a caption, two questions, and a concept. Their task is to read the caption and answer the questions.

**Instructions:**

You will be presented with sentences and questions about them. Your task is to read each sentence carefully and answer two questions:

1. **Does the sentence below discuss ${concept} and not something related to it?**
2. **Can you answer the question based on the sentence alone?**

For each question, select **"Yes"** or **"No"** based on the following guidelines:

- **For Question 1:**
  - Select **"Yes"** if the sentence directly discusses the specified concept and not something related to it.
  - Select **"No"** if the sentence does not discuss the concept directly or discusses something related but not the concept itself.
- **For Question 2:**
  - Select **"Yes"** if you can answer the question based solely on the information provided in the sentence.
  - Select **"No"** if you cannot answer the question based solely on the sentence, or if additional information is required.

Please refer to the examples below for guidance:

*Sentence:* O'Neal - Q RL Helmet - Bicycle helmet
*Does the sentence below discuss a helmet?* **Yes**
*Explanation:* The sentence says it is a bicycle helmet.
*Question:* What type of helmet is depicted in the image (e.g., sports, construction, military)?
*Can you answer the question based on the sentence?* **Yes**
*Explanation:* This is a bicycle helmet, as stated in the sentence.

*Sentence:* Motorcycle Helmet Motocross Helmet cookie cutter set
*Does the sentence below discuss a helmet?* **Yes**
*Explanation:* The helmet is a motorcycle helmet, so we know it's an actual helmet.
*Question:* What color is the helmet?
*Can you answer the question based on the sentence?* **No**
*Explanation:* The sentence doesn't imply the color of the helmet.

*Sentence:* Photo #2 - Cookie & Cookie Monster
*Does the sentence below discuss a cookie?* **Yes**
*Explanation:* The sentence explicitly mentions "Cookie," identifying it as a concept in the sentence.
*Question:* What shape is the cookie?
*Can you answer the question based on the sentence?* **No**
*Explanation:* The sentence does not provide information about the shape of the cookie, only its presence.

Figure 21: 3 out of 10 examples provided to workers as aid to complete their caption filtering task.

## H PROMPTS IN GRADE

### H.1 CONCEPT COLLECTION

To collect a list of diverse concepts, we prompt GPT-4o (OpenAI et al., 2024) with the following:

```
Provide a CSV of 100 unique concepts, like the example below.
concept_id is an enumeration that begins from 0.
Choose concepts that are easy to visually verify for a VQA model.

concept_id,concept
0, an ice cream
1, a cake
2, a suitcase
3, a clock
```

### H.2 PROMPT GENERATION

The following prompt was used to generate common prompts:

```
Please suggest three typical settings for the concept below.
Note that the output should be a list of strings.

Here's an example:
Concept: a cake
Prompts: [
"a cake in a bakery,
"a cake at a birthday party",
"a cake at a swimming pool"
]

Concept: {concept}
```

This one was used to generate uncommon prompts:

```
Please suggest three atypical settings for the concept below.
Note that the output should be a list of strings.

Here's an example:
Concept: a cake
Prompts: [
"a cake on a weight loss clinic,
"a cake at a gym",
"a cake at a swimming pool"
]

Concept: {concept}
```

### H.3 ATTRIBUTE GENERATION

GRADE first analyzes the specific attributes of the concept provided in the prompt, and then generates questions that can be used to count the occurrences of attribute values in images. Below is the prompt we used with GPT-4o.

```
Help me ask questions about images that depict certain concepts.

I will provide you a concept.
Your job is to analyze the concept's typical attributes
```

```
and ask simple questions that can be answered by viewing the image.

Here's an example:

concept:
a cake
attributes:
cakes can be made in different flavors, shapes,
and can have multiple tiers.

questions:
1. Is the cake eaten?
2. Does the cake have multiple tiers?
3. In what flavor is the cake?
4. What is the shape of the cake?
5. Does the cake show any signs of fruit on the outside or
suggest a fruit flavor?

Now that you understand, let's begin.

concept: {c}
```

## H.4 ATTRIBUTE VALUES GENERATION

To generate attribute values $\tilde{\mathcal{V}}_c^a$ for $\tilde{P}_{V|a,c}$, we provide GPT-4o (OpenAI et al., 2024) with a concept, a question, and a prompt. GPT-4o then outputs a list of attribute values that can match the question (attribute). The process is performed for all prompts mentioning the concept. The sets are then unified with similar answers removed (e.g., "motorbike helmets" is removed, because "motorcycle helmets" already exists). The result of the unification is $\tilde{\mathcal{V}}_c^a$.

```
I have a question that is asked about an image.  I will provide you
↪   with the question and a caption of the image.
Your job is to first analyze the description of the image and the
↪   question, then, hypothesize plausible answers that can surface
↪   from viewing the image. Do not write anything other than the
↪   answer.
Then, I need you to list the plausible answers in a list, just like in
↪   the example below. For example,
Caption: a helmet in a bike shop
Question: What type of helmet is depicted in the image?
Plausible answers: ["motorcycle helmets",
            "bicycle helmets",
            "football helmets",
            "construction helmets",
            "military helmets",
            "firefighter helmets",
            "rock climbing helmets",
            "hockey helmets"]
Now your turn.
Caption: {caption}
Question: {question}
Plausible answers:
```

```
I have a question that is asked about an image. I will provide you
↪  with the question and a caption of the image. Your job is to first
↪  carefully read the question and analyze, then hypothesize
↪  plausible answers to the question assuming you could examine the
↪  image (instead, you examine the caption). The answers should be in
↪  a list, as in the example below. Do not write anything other than
↪  the plausible answers.

Example:
Caption: a helmet in a bike shop
Question: What type of helmet is depicted in the image?
Plausible answers: ["motorcycle helmets",
         "bicycle helmets",
         "football helmets",
         "construction helmets",
         "military helmets",
         "firefighter helmets",
         "rock climbing helmets",
         "hockey helmets"]
Now your turn.
Caption: {caption}
Question: {question}
Plausible answers:
```

## H.5 GENERATING ANSWERS

We use GPT-4o to answer the generated questions with $1,000$ as max tokens and temperature 0. We use the Structured Outputs feature (OpenAI, 2024) to map the natural language answers to attribute values in a single step. Our prompt is straightforward:

```
Answer the following question with one of the categories. To come up
↪  with the correct answer, carefully analyze the image and think
↪  step-by-step before providing the final answer.

Question: {question}
Categories:{categories}
Selection:
```