# OpenReview forum: "GRADE: Quantifying Sample Diversity in Text-to-Image Models"
_ICLR.cc/2025/Conference — Submitted to ICLR 2025_

### Official Review · Reviewer_tBnY · 2024-10-27

**Soundness:** 3
**Presentation:** 3
**Contribution:** 3
**Rating:** 6
**Confidence:** 4

**Summary:**

This paper introduces GRADE (Granular Attribute Diversity Evaluation), a new evaluation approach for quantifying sample diversity in T2I models. GRADE addresses the limitations of existing diversity metrics by focusing more on concept-specific attributes such as shape, color or texture, without replying on reference images. The authors perform comprehensive experiments on 12 T2I models, showing consistent default behaviors and limited diversity across models. They attribute this lack of diversity to underspecified captions in training data, which drive models to favor common attributes by default.

**Strengths:**

- This paper is well-written, well-motivated, and well-organized.
- A key strength is that they provide a clear and structured definition of the diversity metric and the systematic approach in Sec. 3. The method estimates distributions with entropy, and allows for a granular and concept-specific evaluation of the diversity.
- The paper links low diversity in generated images to underspecified captions in training data, suggesting that this lack of diversity might come from biases in the data itself. This takeaway on data quality could help improve diversity by addressing such limitations in the training datasets.

**Weaknesses:**

- Since GPT-4o is used to generate prompts, attributes, and attribute values, its lack of version control could affect the consistency of the GRADE framework as GPT-4o evolves. This may alter diversity analysis results, making reproducibility difficult. Changes in GPT-4o’s outputs could lead to inconsistencies when comparing diversity evaluations of T2I models evaluated at different points in time.
- The definition of diversity in this work is focused on specific attribute variations (like shape, color, etc.) and may miss other important aspects of image diversity. The approach relies heavily on the capability LLMs, which could have their own limitations and biases in terms of diversity.
- There's no limitation discussion for example how GRADE might handle more complex or abstract concepts where attributes are less clearly defined. Another part that would be good to include in the future version of the paper is the compositional diversity or relationships between multiple concepts in the same image.

**Questions:**

For the concept-specific attributes, such as shape, color, and texture, is there a predefined set of attributes (other than shape/color/texture), or is it flexible? What range of different attributes is considered, and how consistent is this range across various concepts? Additionally, if you rely on LLMs to automatically generate these attributes, could this introduce new biases into the evaluation process?

---

> ### Author Response · Authors · 2024-11-15
>
> We sincerely thank you for recognizing our paper as well-written, well-motivated, and well-organized. We are pleased that our clear definition of the diversity metric and the systematic approach in Section 3 resonated with you, as well as our analysis linking low diversity to underspecified training captions.
>
> We thank you for your helpful comments. Please see our response below.
>
> **W1: “Since GPT-4o is used to generate prompts, attributes, and attribute values …”**
>
> We agree that because this version of GRADE relies on GPT-4o, it is not straightforward to compare the exact results of this study to other models. However, this is true to any method that uses GPT-4o, and is not directly related to GRADE. It is seamless to change the underlying model to an opensource version that does not change at all.
>
>
> We opted for GPT-4o as it was the most reliable and fastest model we could use to analyse images in scale (we have analysed over 720K images).
>
> The nature of our paper does not call for future replicability of exact scores. In the future, researchers who use GRADE to report diversity scores for their own models, may opt to use an open LLM and VQA for better replicability.
>
> **W2: “The definition of diversity in this work is focused on specific attribute variations …”**
>
> Thank you for pointing this out, we agree that our definition is not all-encompassing and discuss this in the manuscript under the Ethics Statement (lines 540-548).
>
> It is noteworthy that GRADE could be used in addition to other metrics that capture other forms of diversity.
>
> **W3: “There's no limitation discussion for example how GRADE might handle ..”**
>
> Our framework targets only concepts with visually verifiable properties, as we define attributes in Section 3.1.
>
> We do not find that asking questions that can be visually verified about a concept is inherently limited, but the models that are used to test for this concept (like the VQA), could be. We believe that this will change as LLMs and VQA models improve, and remind that GRADE can be modified to work with new underlying models seamlessly. We have clarified this in a dedicated Limitations section.
>
> Thank you very much for the fantastic suggestions! We agree that compositional diversity and relationships between multiple concepts in the same image are great avenues for research. Indeed, we suggest exactly that in the conclusion (lines 533-537). This is a future work direction which we plan to pursue, but believe it is out of scope for this one.
>
> **Q1: “For the concept-specific attributes …”**
>
> GRADE proposes the attributes automatically, so there are many other attributes than just shape/color/texture. We discuss this in section 3.2, in lines 216-226.
>
> Regarding consistency across concepts -- each set of attributes is tailored to the concept, so they are unique.
>
> If you're interested, you can view the attributes of each concept in the supplementary material we submitted.
>
> Regarding LLMs and biases: We agree with this point, and have added it to a Limitations section. We note that all methods incorporate some form of bias, and while we agree it is crucial to acknowledge their existence, no evaluation method is free of bias.

---

> ### Author Response · Authors · 2024-11-23
> **Follow-up**
>
> Thank you for your attention and dedication in evaluating our manuscript. We hope you have had the chance to consider our response from November 16. In it, we clarify that GRADE is limited by its underlying models, and that reproducing results with GRADE is not a hurdle. These clarifications can also be viewed in our revised paper, in a Limitations section we have added. Furthermore, we have clarified that GRADE measures concept-specific attributes, and not all forms of diversity, as originally stated in our Ethics Statement in the paper, as well as addressed other concerns you have stated.
>
>
> Please let us know if there are any additional points or questions you would like us to clarify. We are more than willing to offer further explanations.
>
>
> Thank you,

---

> > ### Comment · Reviewer_tBnY · 2024-11-28
> >
> > Thank you for the response. The authors' replies have addressed my concerns, and I am raising my score based on the clarifications provided.

---

> > > ### Author Response · Authors · 2024-11-29
> > >
> > > Thank you very much for your thoughtful feedback which improved our paper and for revising our score, we are delighted that you appreciate our clarifications.

---

### Official Review · Reviewer_veBF · 2024-11-01

**Soundness:** 3
**Presentation:** 3
**Contribution:** 2
**Rating:** 5
**Confidence:** 4

**Summary:**

This work aims at tackling the open problem of evaluating diversity of text-to-image generative models. The authors propose GRADE, an approach that has two main components: measuring diversity of specific concepts with respect to attributes selected by a language model, and using as a metric the normalized entropy of the distribution of answers given by visual-question answering (VQA) model as metric for diversity. GRADE’s main components were evaluated by human inspection and the efficacy of GPT-4 as a VQA model was assessed via human evaluation. 12 different text-to-image models were compared in light of the introduced metric which showed that most of them perform similarly in terms of diversity (as measured with GRADE).

**Strengths:**

- S1: This work tackles the extremely relevant open research question of evaluating diversity of text-to-images.

&NewLine;

- S2: The proposed approach, GRADE, proposes to measure diversity in terms of images of specific concepts with respect to relevant factors of variation.

&NewLine;

- S3: GRADE is a reference-free metric and doesn’t rely on training any new models to be computed.

**Weaknesses:**

- W1: The proposed metric does not seem to be effective at discriminating between models. It is not clear what is the statistical significance in the difference of scores reported in Table 4. Given the average and standard deviation of GRADE, it seems that confidence intervals for all models would overlap.

&NewLine;


  - W1.1: There is no rigorous validation of the proposed metric and given that it seems GRADE is not able to properly distinguish 12 generative models, it is unclear to me whether the metric is actually capturing diversity as claimed in the manuscript. Experiments considering images with known ground-truth for diversity should be included in order to show that the proposed metric is in fact performing as expected.

&NewLine;


- W2: The authors claim in the conclusion that “Our experiments demonstrate that GRADE is in agreement with human judgements of diversity”, however there is no evidence in the manuscript to support such a strong claim. The human evaluation performed in this work only takes into account the accuracy of GPT-4 on VQA tasks, and it is not even remotely evaluating diversity of generated images. In fact, a major weakness of this submission is that there is no clear evidence that the proposed metric is capturing diversity as perceived by humans.

&NewLine;


- W3: The metric relies on how well the VQA model is able to capture nuances in the differences across different values an attribute can assume. For example, if the diversity of sky images is being measured with respect to the color attribute, it is not clear whether dark blue and navy would be considered as different colors by the VQA model.

&NewLine;


  - W3.1: It is unclear what is the interplay between GRADE and other aspects such as alignment between prompt and generated image, and realism. What happens if the main concept is not actually present in the generated image and/or the generated objects do not look realistic? Would the VQA model be able to answer the question correctly?
&NewLine;


- W4: It is not clear how the approach validity as evaluated in Sections 4-a,b,c  would generalise if other concepts/aspects were taken into account, therefore making it unclear how GRADE performs in case diversity with respect to concepts and attributes of different nature such as people,  ethnicity, skin tone, gender, etc.

&NewLine;



- W5: The manuscript lacks clarity in many parts. Please refer to the "Questions" section of this review for more details.

&NewLine;


- W6: There are several missing references in the text and very relevant related work has not been cited by the authors:
  - Diversity metrics:
     - Friedman, Dan, and Adji Bousso Dieng. "The vendi score: A diversity evaluation metric for machine learning." arXiv preprint arXiv:2210.02410 (2022).
     - Pasarkar, Amey P., and Adji Bousso Dieng. "Cousins of the vendi score: A family of similarity-based diversity metrics for science and machine learning." arXiv preprint arXiv:2310.12952 (2023).
     - Alaa, Ahmed, et al. "How faithful is your synthetic data? sample-level metrics for evaluating and auditing generative models." International Conference on Machine Learning. PMLR, 2022.
   - VQA-based approaches to evaluate generative models:
     - Wiles, Olivia, et al. "Revisiting Text-to-Image Evaluation with Gecko: On Metrics, Prompts, and Human Ratings." arXiv preprint arXiv:2404.16820 (2024).
     - Cho, Jaemin, et al. "Davidsonian scene graph: Improving reliability in fine-grained evaluation for text-image generation." arXiv preprint arXiv:2310.18235 (2023).
     - Lin, Zhiqiu, et al. "Evaluating text-to-visual generation with image-to-text generation." European Conference on Computer Vision. Springer, Cham, 2025.

**Questions:**

- Q1: GRADE assigns maximum diversity to cases where all values of an attribute are equally prevalent in the evaluated images. However, roughly, text-to-image models are usually trained to maximize the likelihood of the "real-world" distribution of natural images. In this case, shouldn't one expect that generated images will reflect the distribution of natural images? If so, does it still make sense to expect that the probability of generating a start-shaped cookie should be equal to the probability of generating a round cookie?

&NewLine;

- Q2: GRADE depends on measuring diversity with respect to specific concepts and attributes, but how many concepts and attributes are "enough" to make sure GRADE is an accurate estimate of diversity?

&NewLine;

- Q3: I am confused by  the equation in lines 156 and 157. Does $p_c^i$ correspond to a single prompt (as indicated in the definition of $\mathcal{P}_c$) or a set of prompts (as it seems to be the case given the sum is over $i$)?

&NewLine;

- Q4: Can the authors kindly provide examples of prompt and concept distributions in Section 3-d? I couldn't understand what exactly they corresponded to even after reading the text a few times.

&NewLine;

- Q5: (Related to W1) Please perform statistical significance tests between model scores and report confidence intervals.

&NewLine;

- Q6: (Related to W3.1) A suggestion to elucidate this point would be assessing GRADE's performance on misaligned or unrealistic images, and checking how/if the metric's reliability was affected.


- Minor comments:
  - Typo:
    - Line: 309: this step **a** second time
  - Equations are missing their respective numbers.

---

> ### Author Response · Authors · 2024-11-15
>
> We thank you for agreeing that our work tackles an extremely relevant open research question, and noting GRADE’s reference-free property to be important as well as its focus on measuring diversity of concepts with respect to relevant factors.
>
> Thank you for the many detailed comments and questions. we respond to them in order. due to response character limits, we respond in two messages
>
> **W1: “The proposed metric does not seem to be effective at discriminating between models. … “**
>
> While we indeed show all models are not diverse, they do differ from each other in diversity. For example, the difference between FLUX-dev and SD-1.4 is very apparent, both in the scores and anecdotally (see Figure 3a and Figure 4).
> Thank you for proposing a statistical significance test, we added them and hope the results alleviate your concerns (see below, more detail in the uploaded revision):
>
> ***Statistical significance:***
>
> We run a two-tailed permutation test with 100K permutations between every unique pair of models (66 in total). Our null hypothesis is that the diversity scores of two models are the same and the p-value is compared to a significance level of 0.05.
>
> ***Results:***
>
> The vast majority of pairs are statistically significant.
> The few which are not are anecdotally also of similar quality. For example, all pair combinations of SD-1.1, SD-1.4, and SD-2.1 are not significant, which is not surprising, seeing as these models for the most part share the same underlying models and training data.
>
> **W1.1: “There is no rigorous validation of the proposed metric … “**
> We respond in W1 and W2 to each claim.
>
>
> In addition, we note that a unique advantage of GRADE is that the diversity score it returns can be traced to specific attributes that lack diversity, which can then be addressed.
>
> **W2: “The authors claim in the conclusion …”**
>
> You are right, our experiments do not measure human _perception_ or _judgement_ of diversity. What we do show is:
>
> * The questions can be answered by examining images of the concepts
> * The attribute values are acceptable variations to each concept
> * The VQA is reliable in identifying the attribute values of concepts approximately > 90% of the time
>
> In other words, we show humans agree that GRADE is _accurate_ in implementing our definition.
>
> Following your observation, we amended the text to reflect this change, as well as state in a Limitations section that our definition of diversity does not directly correspond to human perception of diversity.
>
> While we completely agree with the correction, we would like to point out that (to the best of our knowledge) FID, Recall, and even Vendi Score were not shown to clearly capture diversity as _perceived_ by humans either. _Human perception of diversity is an open question in other fields, such as psychology, our definition of diversity over attribute values is less open to interpretation_.
>
>
>
> **W3: “The metric relies on how well the VQA model is able to capture nuances …”**
> Thank you for clarifying that our text is not clear in this regard.
>
> In general, we unify attribute values that have the same semantic meaning. The example we provide in lines 224-225 is of “circular” and “round”, but your example is equally valid.
>
> **W3.1: “It is unclear what is the interplay between GRADE and other aspects such as alignment between prompt and generated image, and realism …”**
>
> Yes, for the vast majority of cases, the VQA would be able to answer the question correctly.
>
> The VQA is prompted to reply with “None of the above” if the attribute value cannot be extracted from the image, as mentioned in Section 3.2 (lines 228-235). This can happen if the concept is not depicted in the image or the attribute value cannot be inferred. We have not found special issues with unrealistic images (results from prompts “an egg in a zero-gravity environment”). We confirm it responds accurately in such cases and does not hallucinate attribute values, as we show in section 4-b and section 4-c, where humans agree with the VQA on such cases too.
>
>
> **W4: “It is not clear how the approach validity as evaluated …”**
>
> Our study does include people, and GRADE did consider ethnicity, skin tone, and gender properties (which can be viewed in the supplementary material).
>
> Example:
>
> people: we have a "person" concept with a question about gender: "Is the person male or female?"
>
> While GRADE did not ask about ethnicity directly, it did ask about skin tone: For our concept "princess", we have the question "What is the skin color of the princess?".
>
> **W5: The manuscript lacks clarity in many parts. Please refer to the "Questions" section of this review for more details.**
>
> We respond to the questions in a separate comment.
>
> **W6: "... missing references ..."**
> Thank you for these references! We have added all references pertaining to diversity metrics.
>
> We note that while these citations improve the related work section, the takeaway of our paper is not changed.

---

> > ### Author Response · Authors · 2024-11-15
> >
> > **Q1: “GRADE assigns maximum diversity to cases where all values of …”**
> >
> > In our work, we make the distinction between what we expect the model to do (maximize likelihood) and what diverse behavior would look like over concepts and attributes, independent of the training data. We measure the latter, while other metrics, such as FID, measure the former.
> >
> > To illustrate this subtle difference: If all CEOs in the training data are male, the model is expected to consistently generate male CEOs, but that does not mean we would want to mark that model as diverse, since females are left out.
> >
> > Users in a creative setting, for example, may want to use the generation process as a brainstorming tool, and explore unique variations of concepts. Regardless, we find that quantifying this behavior useful, and that since our metric is interpretable about the axes used to measure diversity, users can decide based on their use case.
> >
> > **Q2: “GRADE depends on measuring diversity …”**
> >
> > Thank you for pointing it out. Ultimately, the overall diversity score is indeed conditioned on the set of concepts we used. We opted to use a 100, with 600 images from each concept, but any other number of concepts could have been used, as the results should only be taken in the context of the concepts that were measured.
> > Following this comment, we clarified in a Limitations section that the score is conditioned on the concepts (and attributes) used to measure, and that there is no direct implication on the diversity of concepts that were not measured.
> >
> > **Q3: “I am confused by the equation in lines 156 and 157 …”**
> >
> > Thank you for pointing this out. We have since clarified our notation.
> > $p^{i}_{\text{c}}$ denotes a single prompt, and the sum is over the prompts in the set $\mathcal{P}\_{\text{c}}$, which all mention the concept $c$.  In our paper, as stated in the Method section, the set size is six, but this can be modified.
> >
> > **Q4: “Can the authors kindly provide examples …”**
> >
> > The text in Section 3-d largely repeats the definition in Section 3.1 (lines 152-160 for concept distributions and 189-191 for prompt distributions).
> >
> > Here are some examples:
> >
> > Prompt distribution is the distribution over a specific prompt p (“a car in a car dealership”) that mentions a specific concept c (“car”). You can see examples in Figures 7, 8, and 9.
> >
> > Concept distribution is the distribution resulting from summing all prompt distributions that mention the same concept, over the same question. For example, Figure 7 shows the prompt distribution of the prompt "A car at a car dealership" and the question "What is the color of the car?". The concept distribution of "car" and the question “What is the color of the car?” is obtained by first summing the prompt distributions over the same question and then normalizing the values to cumulatively sum to 1. This is exemplified in lines 155-158.
> >
> > **Q5: “(Related to W1) Please perform …”**
> >
> > Response in W1.
> >
> > **Q6: “(Related to W3.1) A suggestion …”**
> >
> > Response in W3.1.
> >
> > **Minor comments:** addressed.

---

> ### Author Response · Authors · 2024-11-23
> **Follow-up**
>
> Thank you for your attention and dedication in evaluating our manuscript. We hope you have had the chance to consider our response from November 16. In it, we detail a statistical significance test that shows GRADE captures the difference between models, clarify our statement that GRADE reflects human judgement of diversity, and the motivation to measure diversity independent of the training data distribution, as well as address the rest of the concerns and questions. We have also provided a revised manuscript with highlights that mark the changes.
>
>
> Please let us know if there are any additional points or questions you would like us to clarify. We are more than willing to offer further explanations.
>
>
> Thank you,

---

> > ### Comment · Reviewer_veBF · 2024-11-25
> >
> > Dear authors,
> >
> > Thank you for the rebuttal.
> >
> > After reading the comments and the revised version of the paper, I decided to keep my score since my major concerns remain. For example, although the authors performed significance tests as per my suggestion, results are not properly shown in the manuscript (not available in the main results in Table 1), and only briefly mentioned in the Appendix without a proper discussion of why this specific test was chosen and for which pairs of models no significant differences in scores was found.

---

> > > ### Author Response · Authors · 2024-11-25
> > > **Thank you for taking the time**
> > >
> > > Thank you for taking the time to write back!
> > >
> > > **Other concerns you may have had**
> > >
> > > Your initial review includes 8 weaknesses and 6 questions. We have noted your response is about just one of them. Did we improve the paper in regard to the rest of them to your satisfaction?
> > >
> > >
> > >
> > > **Add results to manuscript not available in main results in Table 1**
> > >
> > > Indeed, we refer to the significance tests in Section 5.1 (line 362) in the main results, but do not change Table 4. We will change the first sentence to say “Table 4 presents the mean entropy of models across both multi- and single-prompt distributions, *****with permutation tests showing the results are statistically significant in Appendix D.2*****.”
> > >
> > > *Note: In our revision, concept distribution is renamed to multi-prompt distribution, and prompt distribution to single-prompt distribution.*
> > >
> > > The results of this test do not belong in Table 4, as it shows the average entropy value of each model, which is different from the significance of any two pairs of models. We can include the SEM, which is approximately 0.
> > >
> > > **Only briefly mentioned in appendix**
> > >
> > > Our analysis spans the entirety of appendix D.2, i.e., lines 1123-1145, and not just the four highlighted lines. Apologies if this caused confusion!
> > >
> > >
> > > **No proper discussion why this specific test**
> > >
> > > Permutation tests are the go-to solution in cases of complex data distributions, and are commonly used across science in such cases [1].
> > >
> > > [1] Bonnini, S.; Assegie, G.M.; Trzcinska, K. Review about the Permutation Approach in Hypothesis Testing. Mathematics 2024, 12, 2617. https://doi.org/10.3390/math12172617
> > >
> > >
> > > **No discussion which pairs of models no significant differences in scores was found**
> > >
> > > While we already detail some of the non-significant pairs (lines 1139–1140), and describe broadly the rest (1142–1143), we can include in the appendix every single one of them, if you like.

---

> > > > ### Author Response · Authors · 2024-11-28
> > > > **Thank you again for your response**
> > > >
> > > > Thank you again for your response.
> > > >
> > > > We have revised the paper again following your response, as discussed in our prior comment to you:
> > > >
> > > > 1. Clarified lines 362 to explicitly state the use of permutation tests and the success of the significance test
> > > > 2. Added SEM results to Table 4
> > > > 3. Added to Appendix D.2 the reason we opted for Permutation tests (as well as highlighted the section more clearly)
> > > > 4. As mentioned in our previous comment, in Appendix D.2 we do describe the few models that are not significant.
> > > >
> > > > Please let us know if you have other concerns we can address.

---

> > > > > ### Author Response · Authors · 2024-12-02
> > > > > **Friendly Reminder**
> > > > >
> > > > > As the deadline for reviewers to submit comments is a few hours away, we wish to kindly check in to see if our two responses above sufficiently addressed your concerns. We have provided a comprehensive reply and remain keen to continue the conversation should you have further questions or suggestions.
> > > > >
> > > > > If you feel our response has resolved your concerns effectively, we would be grateful if you could consider improving your rating.
> > > > >
> > > > > Thank you once more for your time and thoughtful consideration.

---

> > > > > > ### Comment · Reviewer_veBF · 2024-12-02
> > > > > >
> > > > > > Thank you for the further clarifications. I updated my score accordingly.

---

> > > > > > > ### Author Response · Authors · 2024-12-04
> > > > > > >
> > > > > > > Thank you for revising your score and for helping us improve the paper!

---

### Official Review · Reviewer_2Lwn · 2024-11-02

**Soundness:** 3
**Presentation:** 3
**Contribution:** 3
**Rating:** 5
**Confidence:** 4

**Summary:**

The paper presents GRADE (Granular Attribute Diversity Evaluation), a novel method for quantifying the diversity of images generated by text-to-image (T2I) models, which often produce limited variations due to underspecified prompts. Unlike traditional metrics like FID, GRADE uses large language models and visual question-answering systems to identify and measure concept-specific attributes, calculating diversity through normalized entropy. The study finds that even the most advanced T2I models frequently exhibit default behaviors, such as generating the same attributes regardless of context, and highlights that biases in training data contribute to these limitations. The authors call for improvements in training data and propose GRADE as a more fine-grained, interpretable diversity metric to advance generative modeling.

**Strengths:**

The strength of this paper is introducing the fine-grained and interpretable metric that overcomes the limitations of traditional diversity metrics by quantifying concept-specific attribute diversity without relying on reference images. It provides deeper insights into text-to-image model behavior and highlights the impact of training data biases, offering a clear path for advancing generative model evaluation and diversity.

**Weaknesses:**

1. Generally, there is a common perception that "cookies" are round, and I share this view. If a square cookie were requested but a round cookie was generated, that would indeed be an issue. However, the problem highlighted by the authors is not about such cases but rather challenges the generalization itself. I find it difficult to relate why generating results that align with common sense is problematic. In other words, it seems the authors are raising an issue with what is an expected outcome.

2. The method uses an LLM to generate questions based on the given prompt, relying heavily on the performance of the LLM. For example, if the prompt involves a cookie, the model generates questions about the shape of the cookie. However, just as many VQA papers specify categories such as shape, attribute, and color, shouldn't there also be a strict specification of categories in this case? If a prompt where shape is crucial fails to address this aspect due to limitations or hallucinations of the LLM, wouldn't that pose a problem?

**Questions:**

1. The distinction between common and uncommon prompts lacks sufficient justification. For instance, in Figure 1, cookies are categorized based on familiar and unfamiliar backgrounds. However, do you think the shape of a cookie should change just because the background is uncommon? Isn’t it more concerning if the object changes rather than the background?

2. How do the authors recognize and address hallucinations when using LLMs and VQA?

---

> ### Author Response · Authors · 2024-11-15
>
> Thank you for your detailed review. We are delighted that you agree that overcoming the reference-based nature of traditional metrics is important, and that our work offers deeper insights into text-to-image model behavior.
>
> **W1: “Generally, there is a common perception that ‘cookies’ are round, …”**
>
> Thank you for bringing this up!
>
> We find that for the most part, it is personal preference: Some people would like to get a diversity of results, and some are happy with a common default. Some would like it for specific properties (eg that generic cookies are round by default) but not for others (eg that generic nurses are female by default). Similarly, users in a creative setting may want to use the generation process as a brainstorming tool, and have it generate non-typical variations of concepts.
>
> Regardless of personal preference regarding this model behavior, it is useful to be able to quantify it, which is the focus of our work. An appealing property of our method is that it is also verbose and transparent about the kinds of diversity (or lack of) it found, so users can decide on a case-by-case basis.
>
> **W2: “The method uses an LLM to generate …”**
>
> By default, GRADE suggests all properties and values automatically but it is straightforward to intervene in the category and values generation step and use other ones.
>
> Because of this modularity, and because the underlying models can be changed with ease, we do not see it as a significant problem.
>
> Thanks to your note, we will clarify in a Limitations section that the diversity score ultimately hinges on the concepts, and data generated by the LLM. However, we remind that we manually validated every set of categories and values that were used in the study, so this limitation is not manifested in our results.
>
> **Q1: “The distinction between common and uncommon prompts …”**
>
> Other tokens in the prompt affect the depiction of the concept we measure. For example, the cookie mentioned in the prompt "a cookie on a table during Christmas festivities" is more likely to take the shape of a tree, or a gingerbread man, because “Christmas” is in the prompt, but it does not represent the typical cookie that one would imagine otherwise.
>
> Common prompts include scenarios which may affect the variation of the concept, as it places the concept in a typical environment. The uncommon prompts purposefully place the concept in uncommon settings to mitigate such common associations
>
> Following your question, we revised section 3.2 paragraph (a) to include the example above.
>
> **Q2: How do the authors recognize and address hallucinations when using LLMs and VQA?**
>
> We measure the reliability of the LLM and VQA models we use (GPT-4o for both) in section 4 and find they are very reliable. However, we acknowledge that they are limited and as any generative model, hallucinate.
>
> We updated the manuscript to clarify that GRADE is limited by the performance of its underlying models, such as the LLM and VQA, as stated in our response to W2.

---

> ### Author Response · Authors · 2024-11-23
> **Follow-up**
>
> Thank you for your attention and dedication in evaluating our manuscript. We hope you have had the chance to consider our response from November 16. In it, we clarify the motivation to measure diversity independent of the training data distribution, and that the underlying models, like the LLM and VQA limit the performance of GRADE. We also address your questions, such as clarifying the need to distinguish between common and uncommon prompts. We have already updated a revision too, where you can view our changes–highlighted in yellow.
>
>
> Please let us know if there are any additional points or questions you would like us to clarify. We are more than willing to offer further explanations.
>
>
> Thank you,

---

> > ### Comment · Reviewer_2Lwn · 2024-11-26
> >
> > Thank you for your response. Regarding W1, you mentioned that some people may raise an issue with generating outputs based on common sense, while others may not. I believe I fall into the latter group; thus, Figure 1 holds no significance for me. I suggest either identifying the real issues caused by uniform generation and replacing the example with those, or moving Figure 2 to an earlier section. Also, could you share how many people participated in the human evaluation?

---

> > > ### Author Response · Authors · 2024-11-26
> > > **Thank you for writing back**
> > >
> > > Thank you very much for your response!
> > >
> > > Figure 1: We are happy to replace Figure 1 with a concept and attribute that are less intuitive based on common sense. What do you think of the example in Figure 3 in the revised PDF (i.e., the color of her dress and the race of the children)?
> > >
> > > Regarding the number of people participated: 71 unique anonymous users sourced by the Amazon Mechanical Turk platform.
> > >
> > > We are happy to answer more questions if you have them!

---

> > > ### Author Response · Authors · 2024-11-28
> > > **Thank you again for your response**
> > >
> > > We have changed Figure 1 to illustrate a less controversial example – the concept “a cup of soda” is consistently mapped to a cup with cola-like fluid with ice, despite that soda applies encompasses many flavors, like water, raspberry, lemonade, etcetera. To be clear, this is not an introduction of a new concept or images, but simply swapping Figure 1 with images of a different concept from our study. This can be verified by viewing the concept in the supplementary data.
> > >
> > > We also added the number of total participants (71) to the details of the human evaluation of GRADE in Appendix G.
> > >
> > > Please let us know if all concerns are resolved.
> > >
> > > Thanks!

---

> > > > ### Comment · Reviewer_2Lwn · 2024-11-29
> > > >
> > > > Thank you for your response. I still have some concerns, and a few issues remain unresolved from the above.
> > > >
> > > > 1. Regarding W2, my question was whether the choice of which attributes to measure depends on the LLM. For some prompts, attributes like "shape" may not be relevant. For example, someone might want to measure the diversity of a cookie's ingredients instead. For the revised Figure 1, which attributes should be measured? Why are they considered absolute attributes? If human intervention is required each time, doesn't this diminish the novelty of the proposed automated metric?
> > > >
> > > > 2. In point 1, the ambiguity around what to measure seems to make the resulting entropy equally ambiguous. It seems that entropy is not an absolute value but is meaningful only when compared to other models. Is that correct? If so, is it fair to compare the diversity of a prompt that measures "shape" with a prompt that does not measure "shape"?
> > > >
> > > > 3. Ambiguity also arises in distinguishing between common and uncommon prompts. Why must their numbers always be equal? Who defines what is "common" or "uncommon"? According to your response, words like "Christmas" could influence the shape of a cookie to be a tree. If that's the case, wouldn't the choice of uncommon prompts allow intentional manipulation of diversity? It seems that all common and uncommon situations should have an equivalent impact on the object being measured for diversity to be fairly evaluated.

---

> ### Author Response · Authors · 2024-11-29
>
> Thank you again for providing us the opportunity to clarify and improve our paper.
>
> **Regarding W2, my question was whether the choice of which attributes to measure depends on the LLM. ...**
>
> This is exactly one of the strengths of GRADE. As we discuss in lines 47-53, the set of relevant attributes is concept-dependent, and requires world knowledge. We use LLMs to automatically identify that set.
>
> While LLMs might not be exhaustive in identifying all relevant aspects, we expect all strong LLMs to be able to identify a good set of semantically-adequate attributes.
>
> **someone might want to measure the diversity of a cookie's ingredients.**
>
> We do not claim to be exhaustive. If someone wants to specifically measure the diversity of a particular attribute they can inject it to the pipeline and run it as-is, however, it is optional.
>
> **For the revised Fig 1, which attributes should be measured? Why are they considered absolute attributes?**
>
> The attributes for “cup of soda” by GPT-4o are shown below. The figure reflects the relevance of these questions nicely, as the images show bias toward specific answers, shown in italic font below.
> 1. Is there a lid on the cup? *No*
> 2. Does the cup have a straw in it? *No strong bias either way*
> 3. If visible, what flavor is the soda (based on its color)? *Cola*
> 4. Is the cup translucent or opaque? *Translucent*
> 5. Can you see ice in the soda? *Yes*
>
>
> Can you clarify what you mean by *absolute attributes*?
>
> Assuming you believe that in the paper we consider the set of attributes for each concept to be encompassing to the diversity of the entire concept, **then this is not the case.** Please refer to us to the lines that led you to believe that, so we can revise it.
>
> We consider each distribution on its own. I.e., we do not imply the results reflect diversity of unseen attributes. Thanks to your review, our revision states so explicitly in the Limitations section (lines 517-518).
>
> Table 4 shows the average diversity score of a model over all distributions of all concepts, as to show the average diversity score over the attributes we did measure, but we do not generalize our findings of a concept’s diversity beyond the attributes we measured.
>
> **If human intervention is required each time, doesn't this diminish the novelty of the proposed automated metric?**
>
> You're correct, but human intervention is not required at all, as we state in the abstract (line 16), in Section 4 (Lines 264-265), and in the Conclusion (line 526).
>
> **In point 1, ... Is that correct?**
>
> No. The normalized entropy is an absolute value within the range [0,1] that indicates how diverse a concept is over a particular attribute (see Section 3.1 for the formulation).
>
> There is no need to compare it to other models at all to interpret the scores, as a score is strictly bounded, with 1 indicating uniform behavior and 0 indicating consistent selection (see lines 195-196).
>
> **If so, is it fair to compare the diversity of a prompt that measures "shape" with a prompt that does not?**
>
> **No, that would not be fair, but we do not do that.** If that was understood from our paper, please direct us to the lines so we can revise it.
>
> Repeating for clarity: We never compare models on the basis of a concept alone, but on a combination of concept and attribute (which must be the same).
>
> **Ambiguity also arises in distinguishing between common and uncommon prompts. Why must their numbers always be equal?**
>
> We kept them equal for our study, to not have an overwhelming representation of a prompt that is inherently biased, like the “Christmas” example, which results in an untypical number of tree cookie shapes, compared to the typical cookie.
>
> **Who defines what is "common" or "uncommon"?**
>
> We determined commonness by checking the frequency of co-occurrences of content words in captions in LAION-5B (Lines 267-287). We found that the average co-occurrence frequency for a common prompt is 30,655 while for an uncommon prompt just 956.
>
> **... It seems that all common and uncommon situations should have an equivalent impact on the object being measured for diversity to be fairly evaluated.**
>
> We agree! The choice of any prompt directly affects the diversity score, this is why in addition to common prompts (which may be biased), we have uncommon prompts, which are less associated with the concept (this is stated in lines 211-223). I.e., uncommon prompts *prevent* manipulation of diversity.
>
> This balance is needed if we want to measure the diversity of the *typical* cookie. If we are to measure the attribute diversity of a concept in a specific setting (e.g., Christmas), we would not need uncommon prompts (which GRADE can accommodate). But as stated in the abstract and introduction, we focus on underspecified concepts (e.g., cookie and not cookies specific to Christmas).
>
> *Thank you again for responding with thoughtful follow up questions, we appreciate it.
> Please let us know if you still have concerns!*

---

> > ### Author Response · Authors · 2024-12-02
> > **Friendly Reminder**
> >
> > As the deadline for reviewers to submit comments is a few hours away, we wish to kindly check in to see if our response has sufficiently addressed your concerns. We have provided a comprehensive reply and remain keen to continue the conversation should you have further questions or suggestions.
> >
> > If you feel our response has resolved your concerns effectively, we would be grateful if you could consider improving your rating.
> >
> > Thank you once more for your time and thoughtful consideration.

---

### Author Response · Authors · 2024-11-15

Dear Reviewers,


We sincerely thank you for your thoughtful and detailed reviews of our manuscript. Your insightful feedback has been invaluable in helping us refine and improve our work.
We have carefully considered all your comments and have made revisions to address your concerns and suggestions. The main changes include:

**Clarifications to the manuscript:**

* Reviewers 2Lwn and veBF sought clarifications about the intuition of measuring diversity independent of the distribution of the training data. We have updated the manuscript in Introduction.
* Reviewer 2Lwn commented that the distinction between uncommon and common prompts is unclear. We have clarified the text accordingly.
* Reviewer veBF pointed out we do not show GRADE is in agreement with human judgements of diversity, but that humans agree GRADE is accurate. We revise the text to reflect that.
* All three reviewers noted there is no discussion on the limitations of GRADE, which we have now added as a section before Conclusion.

**Statistical significance test:**

Reviewer veBF requested we test for statistical significance based on the scores and standard deviations in Table 4. In Appendix D.2 we detail comprehensive permutation tests and find the results to be significant.


In addition to describing the changes in our responses to each of you, we upload an updated manuscript, with the changes highlighted. We believe these revisions have strengthened the manuscript and have clarified any ambiguities. We kindly invite you to review the revised version and hope that it meets your expectations.


Thank you again for your time and constructive feedback.

---

### Meta-Review · Area_Chair_DvMy · 2024-12-18

**Metareview:**

This paper studies the problem of generated image diversity for text to image models. The authors propose to construct a dataset of prompts that asks for objects in common and uncommon scenes. They then use an LLM to generate QA pairs that are answered by a VQA pair to characterize the image diversity on the prompts. The resulting dataset, GRADE, is used to evaluate text to image models. THe key finding in the work is that text to image models consistently generate concepts with same attributes.

Strengths
1. The metric is reference free from images and interpretable.
2. The experiments in this paper look at a variety of different text to image models.

Weaknesses
1. Most modern text to image models, like DALLE-3, SD3, FLUX use an LLM to rewrite the prompt and add details to the text prompt to over-specify object attributes. This technical detail makes a big difference in image quality and diversity and it is unclear if it has been accounted for in the paper.
2. The reporting bias in image captions is a well  reported problem. The authors show that this affects text to image models that are trained on such captions. However, this isn't a surprising claim.
3. Relying on an automated VQA model for evaluation has its own limitations -- the VQA models themselves are biased or limited and thus the evaluation itself is limited. This hasn't been discussed in detail.
4. GRADE is limited to analyzing the objects and attributes in the prompts. This has been stated by the authors too.

Justification
The paper received weak reviews and none of the reviewers judged it to be exciting. The AC concurs with the reviewers about the weaknesses. In particular, the evaluation in this work is limited and the finding isn't novel.

**Additional Comments On Reviewer Discussion:**

The authors noted on Nov 25 that the reviewers not responding to them. However, this comment was not valid and there has since been constructive discussion between the reviewers and the authors.
Two of the reviewers' concerns remain unanswered about the value of the benchmark given its limitations. The third reviewer raised their rating of the paper, but to marginally above acceptance, and isn't a strong champion of this work.

---

### Decision · Program_Chairs · 2025-01-22

Reject